# Interferon stimulated immune profile changes in a humanized mouse model of HBV infection

Yaping Wang[1,7], Liliangzi Guo[1,7], Jingrong Shi [1,7], Jingyun Li[2,7], Yanling Wen[3,7], Guoming Gu[4,7], Jianping Cui[1], Chengqian Feng[1], Mengling Jiang[1], Qinghong Fan[1], Jingyan Tang[1], Sisi Chen[1], Jun Zhang[1], Xiaowen Zheng[1], Meifang Pan[1], Xinnian Li[5], Yanxia Sun[6], Zheng Zhang [3], Xian Li[4], Fengyu Hu[1], Liguo Zhang [2], Xiaoping Tang [1] ✉ & Feng Li [1] ✉

The underlying mechanism of chronic hepatitis B virus (HBV) functional cure by interferon (IFN), especially in patients with low HBsAg and/or young ages, is still unresolved due to the lack of surrogate models. Here, we generate a type I interferon receptor humanized mouse (huIFNAR mouse) through a CRISPR/ Cas9-based knock-in strategy. Then, we demonstrate that human IFN stimulates gene expression profiles in huIFNAR peripheral blood mononuclear cells (PBMCs) are similar to those in human PBMCs, supporting the representativeness of this mouse model for functionally analyzing human IFN in vivo. Next, we reveal the tissue-specific gene expression atlas across multiple organs in response to human IFN treatment; this pattern has not been reported in healthy humans in vivo. Finally, by using the AAV-HBV model, we test the antiviral effects of human interferon. Fifteen weeks of human PEG-IFNα2 treatment significantly reduces HBsAg and HBeAg and even achieves HBsAg seroconversion. We observe that activation of intrahepatic monocytes and effector memory CD8 T cells by human interferon may be critical for HBsAg suppression. Our huIFNAR mouse can authentically respond to human interferon stimulation, providing a platform to study interferon function in vivo. PEG-IFNα2 treatment successfully suppresses intrahepatic HBV replication and achieves HBsAg seroconversion.

Chronic hepatitis B virus (HBV) infection remains a threat to public health, affecting an estimated 250 million people (3.5% of the population) worldwide and causing liver diseases such as hepatitis, fibrosis, cirrhosis and hepatocellular carcinoma (HCC), with approximately 900,000 annual mortalities[1]. As the main approved therapy option, nucleos(t)ide analogs (NAs) targeting the viral reverse transcription step can effectively block progeny virus generation and prevent and even reverse hepatic complications[2]. However, chronic hepatitis B

[1]Institute of infectious Diseases, Guangzhou Eighth People's Hospital, Guangzhou Medical University, 8 Huaying Road, Baiyun District, Guangzhou, Guangdong Province, China. [2]CAS Key Laboratory of Infection and Immunity, Institute of Biophysics, Chinese Academy of Sciences, Beijing, China. [3]Institute for Hepatology, National Clinical Research Center for Infectious Disease, Shenzhen Third People's Hospital; The Second Affiliated Hospital, School of Medicine, Southern University of Science and Technology, Shenzhen, China. [4]Guangzhou XY Biotechnology Co., Ltd, Room 2048, Building 1, No. 6, Nanjiang Second Road, Pearl River Street, Nansha District, Guangzhou, China. [5]Guangzhou Forevergen Medical Laboratory, Room 802, No. 8, Luoxuan 3rd Road, Haizhu, Guangzhou, Guangdong, China. [6]Cytek (Shanghai) Biosciences Co, Ltd, Guangzhou, China. [7]These authors contributed equally: Yaping Wang, Liliangzi Guo, Jingrong Shi, Jingyun Li, Yanling Wen, Guoming Gu. ✉e-mail: tangxp@gzhmu.edu.cn; gz8h_lifeng@126.com

(CHB) patients, despite achieving undetectable serum HBV-DNA levels, still have a high risk of developing HCC[3–5]. Although serum HBV DNA may be undetectable, the persistent intrahepatic existence of viral mRNAs and proteins (HBcAg, HBx and HBsAg) continually stresses virus-containing hepatocytes and creates an immune tolerance microenvironment in the liver, which eventually leads to hepatocyte malignancy and progression to HCC after decades[6,7].

During the past decade, IFNα treatment has effectively achieved functional cure in some CHB populations (functional cure definition: sustained undetectable HBsAg and HBV DNA in the serum long term after drug cessation). First, over 35% of CHB patients (947 of 2597 in a real-world observation study) with HBsAg below 1500 IU/ml after a certain duration of NAs treatment were reported to achieve functional cure after IFNα treatment[8]. Twelve of 16 (75%) CHB patients with HBsAg <100 IU/ml achieved HBsAg seroconversion, suggesting that the lower the HBsAg levels were, the more effective IFN was in suppressing HBV[9]. Second, CHB patients with intrahepatic inflammation (elevated ALT levels at baseline or after treatment) were reported to be more likely to achieve HBsAg clearance or seroconversion after IFNα treatment[10,11]. Finally, together with other groups, we observed that children with CHB are more responsive to IFNα treatment and achieve undetectable HBsAg more frequently than adult patients with CHB[12,13]. Interestingly, children with CHB (3–7 years old) who quickly achieved undetectable HBsAg also generated high levels of HBsAb, a status similar to that observed after vaccination[12], and early initiation of antiviral therapy for infantile-onset hepatitis B (<12 months old) contributes to a rapid and significant loss of HBsAg[14], indicating an association between HBV infection length and IFNα effectiveness.

There has been an explosion of novel anti-HBV drugs targeting HBV entry, viral capsid assembly, viral translation and secretion and host factors (reviewed in ref. 15). However, increasing numbers of clinical trials have shown that single drug regimens without interferon fail to achieve sustained viral suppression, implying the vital role of immune status reversal in HBV therapy. Indeed, the inclusion of IFNα in HBV therapeutic vaccines broke virus-specific immune tolerance and increased HBsAg seroconversion in a recent study[16]. The "IFNα-plus" strategy that combines IFNα with direct HBV antivirals, in sequence or simultaneously, seems very likely to achieve sustained HBV functional cure. In cell culture models, the IFN pathway has been reported to orchestrate a multipronged attack on HBV virus replication by inhibiting different steps of the viral life cycle, such as viral entry, transcription, translation, genome replication, assembly, and egress, by inducing the expression of several hundred IFN-stimulated genes[17]. Unfortunately, the underlying mechanism in vivo is still elusive.

The interferon family, which consists of type I (human 17 and mouse 14), II (1) and III (human 4 and mouse 2) interferons, is critical for initiating a broad spectrum of immune responses upon infections[18]. On one hand, the species specificity of interferons and their receptors restricts the direct application of human IFNs in wildtype mice because the immune response is not representative. On the other hand, the binding specificity and affinity of interferons result in varied potencies[19,20]. Here, to investigate the function of human IFN in vivo, we generated a type I IFN receptor humanized (huIFNAR) mouse model and characterized gene expression profile changes in response to human IFNα in vivo. We observed that PEG-IFNα2 treatment led to a significant viral HBsAg reduction and extensive immune cell alterations in the AAV-HBV mice.

## Results

### Generation of a humanized type I interferon receptor mouse model

Due to species specificity, mouse interferons (IFNs), which can mimic human IFNs in stimulating identical downstream signaling cascades, are unavailable. The varied binding affinities of IFNs to their receptors due to minor sequence variations (Supplementary Fig. 1a, b) elicit different antiviral effects[19–21]. HuIFNα2, the only family member approved for treating CHB patients, has no identical counterpart in mice (Supplementary Fig. 1c). In addition, the weak similarity of the extracellular domain of the IFN receptor (IFNAR) where IFN binds between humans and mice (58.1% and 50.2% for the interferon receptor 1 (IFNAR1) and IFNAR2 subunit, respectively) hinders attempts to directly apply human IFN in mouse experiments in vivo (Supplementary Fig. 2a, b). We stimulated the PBMCs from wildtype mouse C57BL/6J with human IFNα2 and mouse IFNα5, respectively (Supplementary Fig. 3a). As expected, human IFNα2 failed to stimulate a gene expression profile comparable to mouse IFNα5 (Supplementary Fig. 3b), highlighting the necessity to generate a surrogate mouse model to study the function of human IFNs in vivo.

To investigate human IFN in mice, we designed a tandem expression cassette consisting of both human IFNAR2 and IFNAR1, which were linked by a 2A sequence, and placed its expression under the control of the mouse *Ifnar2* promoter (Fig. 1a, plasmid sequence is in the supplementary Data 1). This design avoided the need for two consecutive rounds of gene manipulation and saved time because the two mouse IFNAR1 and IFNAR2 genes are located close to each other on chromosome 16 (only 130 kb apart). For the mouse *Ifnar2* gene, insertion of a foreign sequence in exon 2 next to the ATG start codon not only retained the promoter activity undisturbed but also retained an intact intron between exon 1 and exon 2, increasing the stability of the novel mRNA. Finally, a polyA signal sequence at the 3′ end provided a transcription stop signal and aborted mouse IFNAR2 expression. Therefore, our design maintained the bone fide gene expression of the inserted human IFNAR1 and IFNAR2 genes while causing mouse IFNAR defects.

In the chimeric huIFNAR, the extracellular human moiety could faithfully respond to human IFNα stimulation, while the intracellular mouse moiety could guarantee binding protein recruitment and the downstream pathway activation (Fig. 1b). Homozygous human interferon receptor humanized (huIFNAR) mice generated using CRISPR/Cas9-aided pronuclear microinjection were confirmed using PCR (Supplementary Fig. 4a, b) and gene sequencing (in the Supplementary Data 2). HuIFNAR mRNA expression was confirmed using reverse transcription PCR (Fig. 1c). Normal expression of the huIFNAR2 protein on the cell surface was detected by flow cytometry (Fig. 1d). Finally, whether the huIFNAR mouse could respond to human type I interferon injection was tested in vitro and in vivo. Mouse PBMCs were treated with human IFNα2 with or without the human IFNAR1 and IFNAR2 blocking antibodies (anti-hR1 and anti-hR2, respectively, Fig. 1e). Human IFNα2 treatment resulted in upregulated levels of mouse Mx1 (mMx1) and mouse Isg15 (mIsg15) mRNA (Fig. 1f). Nevertheless, this activation effect was diminished when huIFNAR was blocked by anti-hR1 and/or anti-hR2 antibodies (Fig. 1g), implying that increased mMx1 and mIsg15 expression induced by huIFNα2 was mediated by humanized type I IFN receptors. To test the response of huIFNAR mouse in vivo, intraperitoneal injection of human PEG-IFNα2 was done as indicated (Fig. 1h). Considerably elevated mMx1 and mIsg15 mRNA levels were observed in PBMCs, the liver and the spleen in huIFNAR mice (Fig. 1i, j), further confirming that the huIFNAR mouse was successfully constructed and could respond to human interferon stimulation. In addition, we measured the mISG15 expression in the spleen tissue by western blot (Supplementary Fig. 5a). Human PEG-IFNα2 elicited a marked increase of mISG15 in the protein level compared with the mouse IFNα5 (Supplementary Fig. 5b).

Given the species specificity of the type I interferon pathway, we hypothesized that the huIFNAR mouse would have an altered response to human IFN stimulation, compared with the wild-type C57BL/6J to the mouse IFNs. To test this hypothesis, we compared

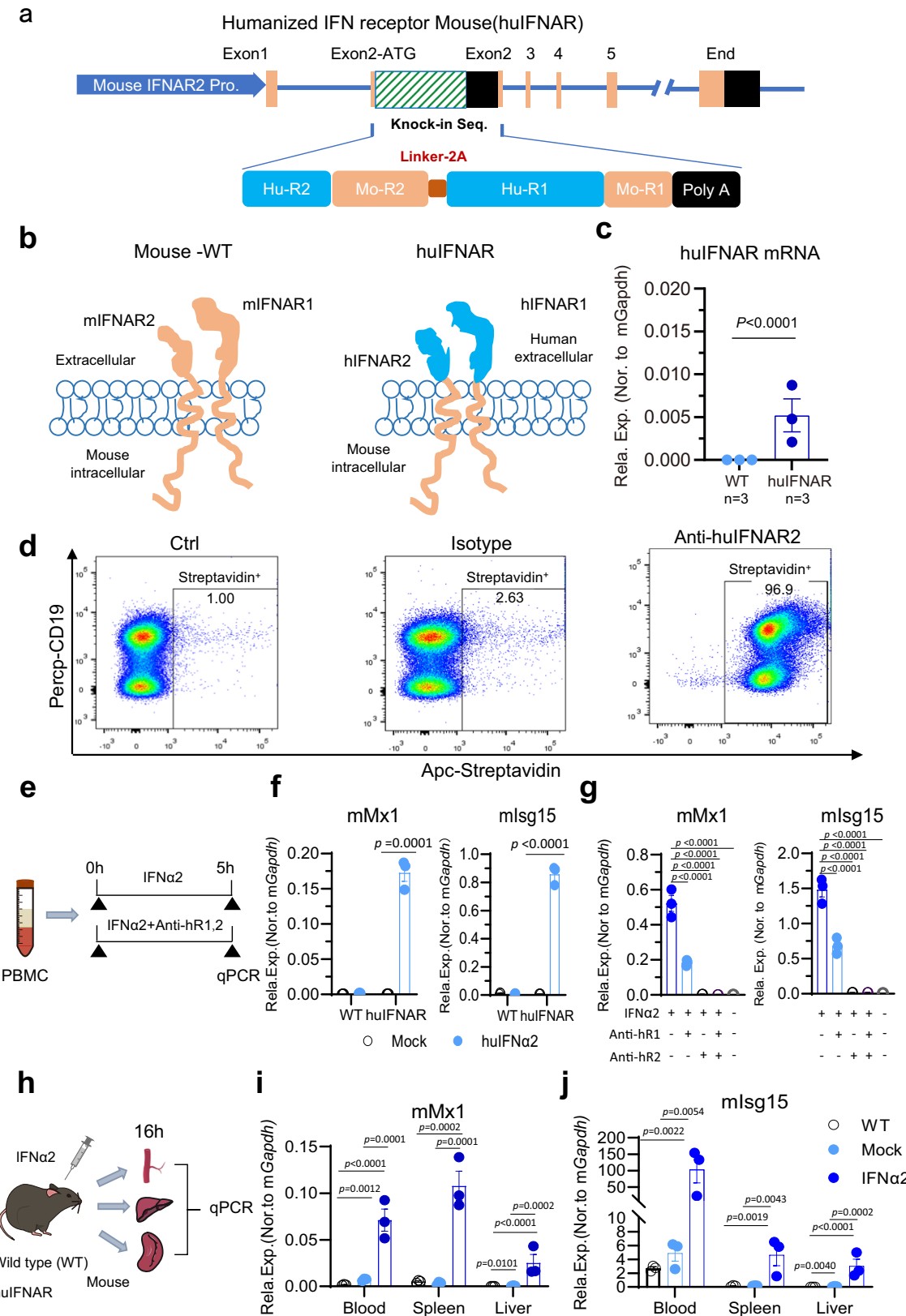

the gene expression profiles against interferon stimulation in the wildtype and huIFNAR mouse. PBMCs isolated from wildtype C57BL/6J and huIFNAR mice were stimulated with huIFNα2 and mouse IFNα5 for 8 h (short-term) and 20 h (long-term), respectively, and subjected to a transcriptomic analysis of the gene expression (Supplementary Fig. 6a). As expected, the heatmap, containing all

the differently expressed genes at different time points post IFN stimulation, showed obvious gene expression differences between the mIFNα5 and huIFNα2 (Supplementary Fig. 6b), confirming that the huIFNAR mouse altered its type I interferon pathway. Taken together, we successfully generated functional type I interferon receptor-humanized mice.

**Fig. 1 | Generation and functional evaluation of a type I Interferon receptor-humanized (huIFNAR) mouse. a** Schematic diagram of huIFNAR knock-in strategy in the exon 2 of mouse Ifnαr2 gene. The upper, from left to right, consists of mouse Ifnαr2 promoter (deep blue), Exons (orange), the insert, and end. The chimeric receptor structure (below) is indicated. Hu: human; Mo: Mouse. R2: interferon receptor 2, R1: interferon receptor 1. Linker-2A: linker region and 2A self-cutting protease. PolyA: transcription termination signal. **b** Topological structure of humanized interferon receptor. Blue, human origin; orange, mouse origin. **c** Relative huIFNAR mRNA expression to mouse Gapdh (mGapdh) by reverse transcription quantitative PCR (qPCR), $n = 3$ biologically independent WT mice and huIFNAR mice, respectively. **d** Flow cytometry determination of huIFNAR protein on the cell surface. **e**–**g** huIFNAR response to human IFN stimulation in vitro. Mouse PBMCs isolated from blood was stimulated for 5 h ex vivo with human IFNα2 (800 ng/ml) or plus huIFNAR blocking antibodies. anti-hR1 and anti-hR2 represent anti-human IFNAR1 and IFNAR2 antibodies (10 μg/ml), respectively. The mMx1 and mIsg15 normalized to mGapdh or mActin are indicated. $n = 3$ biologically independent mice in each group. **h**–**j** Functional assess of huIFNAR response to PEG-IFNα2 (2 μg/ml for 16 h, subcutaneous injection) in vivo. The relative levels of mMx1 and mIsg15 mRNA to mGapdh in blood PBMC, liver and spleen are indicated. For two group comparison, continuous variables were represented as mean ± SEM or median and interquartile range (IQR), and compared by two-sided unpaired t-test or Mann–Whitney $U$ test as appropriate. For multiple comparison, ANOVA test or Kruskal–Wallis test was applied. $n = 3$ biologically independent mice in each group. Source data are provided as a Source Data file.

## Peg-IFNα2 stimulates similar immune responses in the huIFNAR mouse PBMCs to those in human PBMCs

To further confirm whether these mice could represent humans in terms of the response to huIFNα, the stimulated gene expression profiles in mouse PBMCs and human PBMCs were compared using mRNA sequencing (Fig. 2a). First, 199 of total 487 ISGs had significant changes after huIFNα treatment compared with the mock treatment. The IFN-stimulated gene (ISG) expression profiles between humans and mice were found to be well matched (Fig. 2b). Next, the similarity of differentially expressed genes between the huIFNAR mice and humans was assessed. The top 5000 differentially expressed genes from each huIFNAR PBMCs and human PBMCs were employed to enriched gene pathway analysis (Kyoto Encyclopedia of Genes and Genomes, KEGG). PEG-IFNα stimulated changes in 109 enriched KEGG pathways in huIFNAR PBMCs and 146 in human PBMCs, among which 92 enriched KEGG pathways were shared (56.7% of the total changed pathways) (Fig. 2c). We chose two representative KEGG pathway to show their similarity. Multiple shared nodes in the NOD-like receptor signal pathway were upregulated after PEG-IFNα2 treatment in both huIFNAR mouse and human PBMCs (Supplementary Fig. 7a). Similarly, the JAK-STAT signaling pathway was also synchronously triggered (Fig. S7b). Beyond ISGs, we performed a comprehensive gene ontology (GO) analysis of biological processes (BPs) for all DEGs. In line with the KEGG Orthology-Based Annotation System (KOBAS), the gene ratios (the numbers of regulated genes stimulated by Peg-IFNα2 versus the total number of genes from the same annotation) between the two groups were very similar (Supplementary Data 3). The top 30 most significantly enriched BPs were very similar between huIFNAR mice and humans (Fig. 2d). Changes related to the immune system process (the top GO:0002376 in Supplementary Data 3) and metabolism (a nonimmune-related GO:0008152 in Supplementary Data 3) further confirmed this similarity (Fig. 2e, f). Altogether, the high similarity of gene responses between huIFNAR mouse and human PBMCs induced by human Peg-IFNα2 indicated that the huIFNAR mouse could be employed as an alternative model to explore the function of human type I interferons in vivo.

## The atlas of the tissue-specific response to human IFNα2

A comprehensive investigation of the human tissue-specific response to IFN treatment in healthy individuals is not possible in humans but became feasible with the huIFNAR mouse. Here, nine main tissues, namely, the brain, blood, lung, heart, liver, spleen, kidney, muscle and intestine, were chosen to depicture the tissue-specific response to human PEG-IFNα2 stimulation in vivo (Fig. 3a). Varying levels of huIFNAR mRNA were observed among different tissues (Fig. 3b). The blood, liver and spleen expressed the highest levels of huIFNAR. In contrast, the muscle, intestine and brain expressed the lowest levels. Accordingly, the response to IFNα2 varied substantially across tissues (Fig. 3c). The blood (5102 genes), liver (3805 genes) and spleen (3719 genes) exhibited the most obvious profile changes in response to PEG-IFNα2 stimulation. Meanwhile, the heart (124 genes), brain (168 genes) and intestine (413 genes) showed only a weak response. The tissue-specific gene expression changes were confirmed by Q-PCR analysis (Supplementary Fig. 8).

In clinical practice, blood samples are generally utilized as surrogates to surveil and predict disease outcomes due to the limited availability of other tissues. However, the representativeness and validity of blood tests have been questioned. Here, we observed that in the differentially expressed gene profiles among the blood, liver and spleen (Fig. 3d), only 629 genes were shared, 2855 genes were unique to the blood, 1884 were unique to the liver, and 3719 were unique to the spleen. More genes showed differential expression in the blood (5102 genes) than in the liver (3805 genes), and only 1352 genes were shared by the blood and liver. Hierarchical clustering revealed that a small gene cluster was significantly upregulated in the liver compared with the blood (*Creb5, Apol10b, Ddx4, Siglec1, Il1rn, Irgbzb1, Ifit2, Oas3, Cxcl9, Gbp11, Ifi214, Ms4a4c, Phf11, Phf11b, Phf11d,* and *Olfr56*, labeled red, Fig. 3c). Another gene cluster was downregulated in the blood compared with the liver (*Angptl8, Saa1, Saa2, Fqb, Pck1, Hpx, Serpina3m, Serpina3n, Orm1, Ube2l6, Apob, Rrares2,* and *Rbp4,* labeled blue, Fig. 3c). In addition, our analysis of the alterations of the top 50 ISGs revealed stark differences among tissues in terms of number and magnitude, highlighting that each tissue has a unique sensitivity to interferons (Fig. 3e).

ISG expression alteration will inevitably result in a cascade of a broader gene expression. After categorizing all the DE genes (Fig. 3f), we found that GO terms, including response to interferon-gamma and interferon-beta, cellular response to interferon-beta, defense response to virus, response to virus, symbiotic process, positive regulation of response to external stimulus, positive regulation of cytokine production, regulation of response to biotic stimulus, and negative regulation of immune system process, were all enriched in all the tissues. In the liver, spleen and lung, we observed GO terms related to metabolism (carboxylic acid, organic acid and small molecule catabolic process). Despite its lower level of the huIFNAR mRNA (Fig. 3b), the muscle still had a broader response than the heart, intestine or brain. The comprehensive tissue-specific gene expression atlas in vivo revealed a previously unappreciated functional difference in the response to human IFNα2 stimulation.

## Human Peg-IFNα2 treatment reduced HBsAg levels in HuIFNAR mice

The huIFNAR mice were then used to evaluate the efficacy of human interferon in treating HBV. An adeno-associated viral vector containing a 1.3-fold HBV genome (AAV-1.3XHBV) was chosen for long HBV persistence in the mice[22]. Of note, we preferred genotype C HBV, which infects the most population (26% of total CHB patients) and is prevalent in the Western Pacific region[23], for the following experiments. Typically, viral HBsAg reached 1000–5000 IU/ml within 7–10 days after tail vein injection of $2 \times 10^{11}$ viral genomes (vg)/mouse, followed by a dramatic decline at weeks 2–3 and a rebound to over 2000 IU/ml then after, and the HBsAg levels stably maintained after week 6 (Supplementary Fig. 9). The HBsAg kinetics mimicked early acute infection, subsequent viral suppression by the host and finally chronic

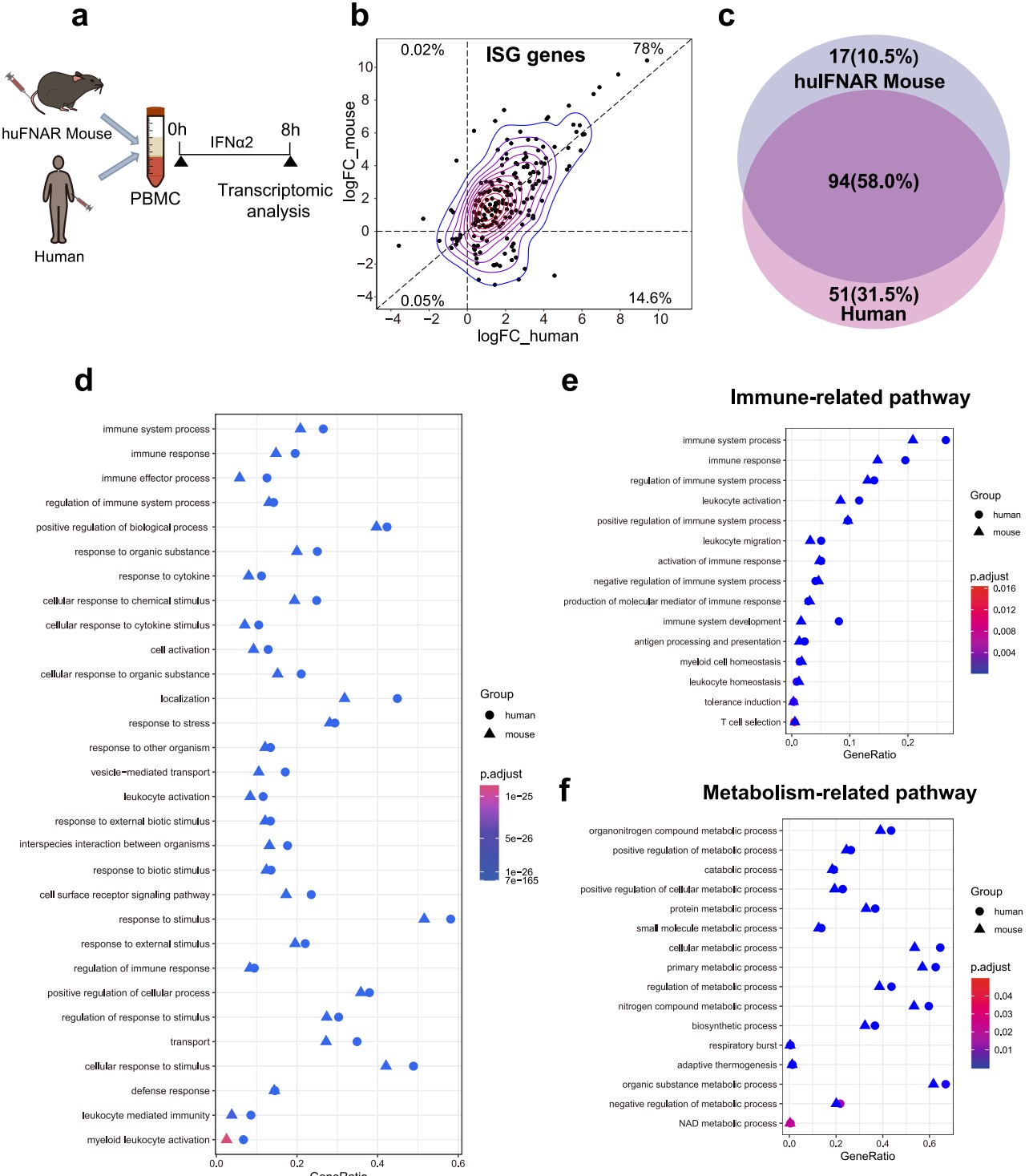

**Fig. 2 | Similarity analysis between the mouse huIFNAR and human PBMCs in response to human IFN stimulation. a** Experiment protocol for comparing the immune response of PBMCs to PEG-IFNa2 (0.8 μg/ml for 8 h) stimulation between health humans (*n* = 3) and huIFNAR mice (*n* = 3). The gene expression profile was analyzed by next-generation sequencing. **b** Correlation of differentially expressed ISGs between human and huIFNAR mice. Each dot represents one gene. Number and percentage are indicated in each quadrant. **c** Venn diagrams of shared enriched KEGG pathways induced by IFN between human and huIFNAR mouse. Number and percentage are indicated. **d** The top 30 most enriched GO terms in all biological processes (BP). Circle, human. Triangle, huIFNAR mouse. **e**, **f** The immune and metabolic-related sub GO terms as in (**d**). Adjusted *p* values in (**d**–**f**) are calculated using two-sided hypergeometric distribution and adjusted by Benjamini–Hochberg method. Source data are provided as a Source Data file.

persistence in humans. The HBsAg kinetics showed in a typical manner to the course of chronic infection in human[24]. HBV DNA showed a similar trend to HBsAg but was stably maintained at approximately 10^6 copies/ml for the genotype C virus. The HBeAg level increased gradually. Genotype C HBV showed higher HBsAg, HBeAg and HBV-DNA levels than the genotype B virus.

AAV-1.3 × HBV (2.0 × 10^11 viral genomes/mouse) was injected via the tail vein 6 weeks before PEG-IFNα2 treatment (2 μg/mouse, once

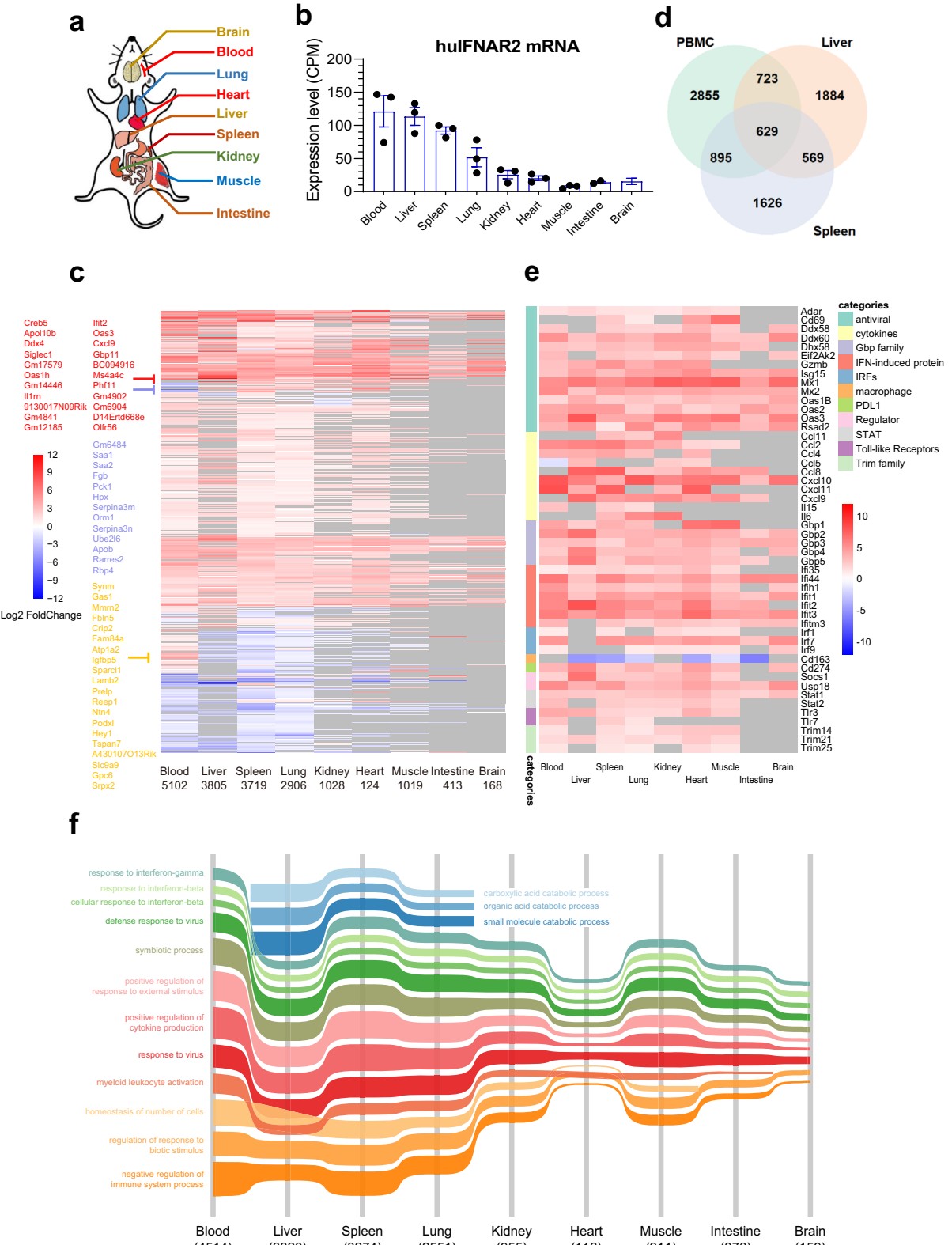

**Fig. 3 | The tissue-specific transcriptome atlas of response to the human PEG-IFNα2. a** Schematic diagram tissue collection as indicated at 16 h after subcutaneous inoculation of human PEG-IFNα2 (2 μg/mouse, in 200 μl buffer). **b** The huIFNAR2 mRNA levels in 9 tissues. *n* = 3 biologically independent huIFNAR mice. CPM counts per million reads. Data are presented as mean values ± SEM. **c** Heatmap of differential expression genes (DEGs). DEGs were defined as significance under an FDR threshold of 0.05 in the IFN group versus the control group. The DEGs numbers were indicated under each tissue. **d** Venn diagrams of the DEGs among PBMC, liver, and spleen. **e** Heatmap of differentially expressed ISGs. **f** Alluvial plot of Tissue-specific GO analysis. A total of 15 GO terms from the top 5 significant biological process terms in each organ was shown. The stream wideness represents the size of annotated DEGs enriched in the GO term. Source data are provided as a Source Data file.

per week) (Fig. 4a). Wildtype C57BL/6J served as the PEG-IFNα2 treatment control. Viral HBsAg, HBeAg, and HBV DNA were measured weekly. Compared with the wildtype mice, HBV biomarkers such as HBsAg and HBeAg differed at early time points (Supplementary Fig. 10).

As expected, PEG-IFNα2 failed to suppress HBV in C57BL/6J mice (Fig. 4b, d). In contrast, the HBsAg in the IFNα-treated group started to decline from week 7 ($p = 0.0112$). No further decline was observed when PEG-IFNα2 was applied twice per week from week 10 until termination (Fig. 4b). The average serum HBsAg level in the PEG-IFNα2 was approximately 8.04-fold lower than that in the mock mice ($p = 0.0001$) (Fig. 4c) at the termination. Similarly, the HBeAg decline started from week 9, but there was only a 2.15-fold reduction at termination ($p = 0.0069$) (Fig. 4d, e). One PEG-IFNα2-treated huIFNAR mouse (#3) achieved HBsAg and HBeAg loss (Fig. 4f), and the other treated huIFNAR mouse had a marked HBsAg reduction after receiving PEG-IFNα2 treatment (Supplementary Fig. 11). Surprisingly, HBsAb was detected in two of six mice in the PEG-IFNα2 group (Fig. 4g). The intrahepatic viral pgRNA and total mRNA were reduced by PEG-IFNα2 and were nearly completely inhibited in the #3 mouse (Fig. 4h, i).

Unfortunately, no significant HBV DNA decrease was observed after IFNα treatment (Supplementary Fig. 12). We noticed that the serum HBV-DNA level was approximately two log10 lower in the huIFNAR mouse than in the wildtype mouse (Supplementary Fig. 12). Intrahepatic viral rcDNA revealed that the HBV rcDNA level in the huIFNAR model was half a log10 (approximately threefold) lower than that in the wildtype C57BL/6J AAV-HBV model (Fig. 4j). The underlying mechanism warrants further investigation.

We also analyze the possible side-effector of long-term IFN treatment. The liver tissue was stained with hematoxylin and eosin. We could not find abnormality compared with the mock-treated huIFNAR mice and wildtype C57BL/6J mice (Supplementary Fig. 13). Meanwhile, we further measured the kinetics of proinflammatory cytokines, consisting of mouse IL-23, IL-1α, IFN-γ, TNF-α, CCL2 (MCP-1), IL-12p70, IL-1β, IL-10, IL-6, IL-27, IL-17A, IFN-β and GM-CSF. No significant difference between these three groups (Supplementary Fig. 14). Interestingly, the #3 mouse with HBsAg seroconversion showed a trend to have high IL-23, IFN-γ, IL12p70, IL-1β, IL-6, IL-17a and GM-CSF. However, a large cohort is required to study the underlying mechanism.

## Intrahepatic immune cell population alterations after 15 weeks of PEG-IFNα2 treatment

Human IFNα2 seems to inhibit HBV replication in multiple ways[17,25–27]. Generally, the intrahepatic microenvironment alteration is postulated to be vital for HBV functional cure in vivo, and the characteristics of the microenvironment at the termination would be informative. Immune cells isolated from the liver tissues of mock- and PEG-IFNα2-treated huIFNAR mice were subjected to single-cell sequencing (Supplementary Fig. 15a). Due to insufficient cells numbers, intrahepatic immune cells isolated from 2–3 mice in each group were pooled. We compiled gene expression data from 8856 cells for clustering analysis and revealed eight main distinct large cell clusters visualized as a uniform manifold approximation and projection (UMAP) embedding, consisting of lymphoid cells such as B cells, T cells, CD8+ T cells, CD4+ T cells, regulatory T and natural killer (NK) cells, and myeloid cells such as neutrophils and macrophages (Supplementary Fig. 15b–e). PEG-IFNα2 treatment showed a tendency to increase the size of myeloid and neutrophil populations and reduced the size of the NK/T and B cell populations (Supplementary Fig. 15d).

The myeloid cell cluster was further separated into Utg1a-high monocytes, MHC1-high monocytes, Trem1-high monocytes, MKi67+ monocytes (expressing high Ki-67), macrophages, pDC and DCs (Fig. 5a and Supplementary Fig. 16). The population nomenclature was designed according to specific gene expression patterns. IFNα2 treatment showed a tendency to increase the numbers of MHC1-high

monocytes, Trem1 high monocytes and MKi67+ monocytes but reduced the numbers of Utg1a-high monocytes, macrophages, DCs and pDCs (Fig. 5b). Compared to the mock mice, 67 genes were upregulated and 16 genes were downregulated in monocytes from PEG-IFNα2-treated mice (Fig. 5c). These differentially expressed genes were enriched in the NF-kappaB signaling pathway, Toll-like receptor signaling pathway, apoptosis and IL-17 signaling pathways (KEGG) (Fig. 5d). The monocytes seemed to have a proinflammatory profile (cell differentiation, inflammatory response and enhanced cytokine production). In contrast to mock treatment, IFNα2 sensitized the monocytes to be more responsive to IFN, enhanced antigen presentation, elevated co-stimulation molecule expression, more attractive to immune cells, and proliferative activity (Fig. 5e–j). Specifically, the Utg1a-high cluster was more prone to respond to IFN, while the MHCI-high cluster showed a higher antigen presentation score. The Trem1-high cluster seemed less responsive to IFN, with high scores for chemokines pro-inflammation and proliferation (Supplementary Fig. 17). Flow cytometry analysis (Supplementary Fig. 18) revealed substantially increased MHCII+Ly6C+ monocytes and proinflammatory macrophages in the liver after PEG-IFNα2 treatment (Fig. 6a–d).

The NK/T lymphoid lineage cluster constituted the largest population in the liver (Supplementary Fig. 15d). Compared with the mock mice, we observed that NK (CD56+) and NKT (CD3+CD56+) cells showed a slight decrease in numbers in the PEG-IFNα2-treated mice, but the number of liver resident NK cells (rNK cells, CD49a+CD56+) decreased significantly in the liver (Supplementary Fig. 19). T cells were enriched in the liver, but CD4 levels were unaffected. Therefore, we focused on CD8+ T cells in the following analysis. The CD8+ T cells in the PEG-IFNα2-treated group showed upregulated functions, such as leukocyte cell adhesion, activation and T-cellular receptor signaling (Supplementary Fig. 20). Whole CD8+ cells had higher scores for activation, chemokine, and cytotoxicity and exhaustion. Specifically, we observed an obvious decrease in the Lef1+ CD8 T-cell population but a surge in the Teff CD8 T-cell population (Fig. 7a, b). The markedly changed genes in Teff CD8 cells were mainly enriched in cell differentiation regulation, chemotaxis and chemokine signaling, interleukin-12 production, T-cell receptor and Toll-like receptor signaling clusters (Fig. 7c, d). Hence, intrahepatic Teff CD8 T cells exhibited increased function of activation, cytotoxicity and chemotaxis (Fig. 7e–h). Unexpectedly, we observed that Teff CD8 T cells also dysregulated expression of exhaustion-specific genes (such as *Lag3*, *Havcr2*, *Pdcd1* and *CTLA4*) and exhibited exhausted gene expression profiles, indicating that the effector function of these cell populations included the upregulation of exhaustion molecules. Our flow cytometry analysis also revealed a significantly increased effector memory CD8+ T (TEM) cells (Fig. 8a, b, Supplementary Fig. 18) and both the CD8+ TEM and total CD8 T cells expressed elevated levels of PD-1 (Fig. 8c–f). Finally, we analyzed the HBV-specific T cells immune response using the enzyme-linked immunosorbent spot (ELISpot) assay. Intrahepatic lymphocytes were stimulated with HBV core and HBsAg peptide pool. Secreted mouse IFN-γ was measured. Unfortunately, HBV-specific T cells were not observed (Supplementary Fig. 21). Altogether, our observation indicated that the co-expression of exhaustion biomarkers in effector CD8 T cells might reduce the potency of these cells in inhibiting HBV in vivo.

## Discussion

In this work, we described a humanized mouse model in which human IFNARs (huIFNAR1 and huIFNAR2) were knocked into the mouse *Ifnar2* locus recapitulates the functional response to human interferon stimulation in humans. Although nonhuman primates, treeshrews, and woodchuck mice are susceptible to HBV infection[28], they are rarely used to test human interferons because of their unmatched species-specific interferon receptors between these species and human IFNs. A liver-humanized mouse model with a human primary hepatocyte-

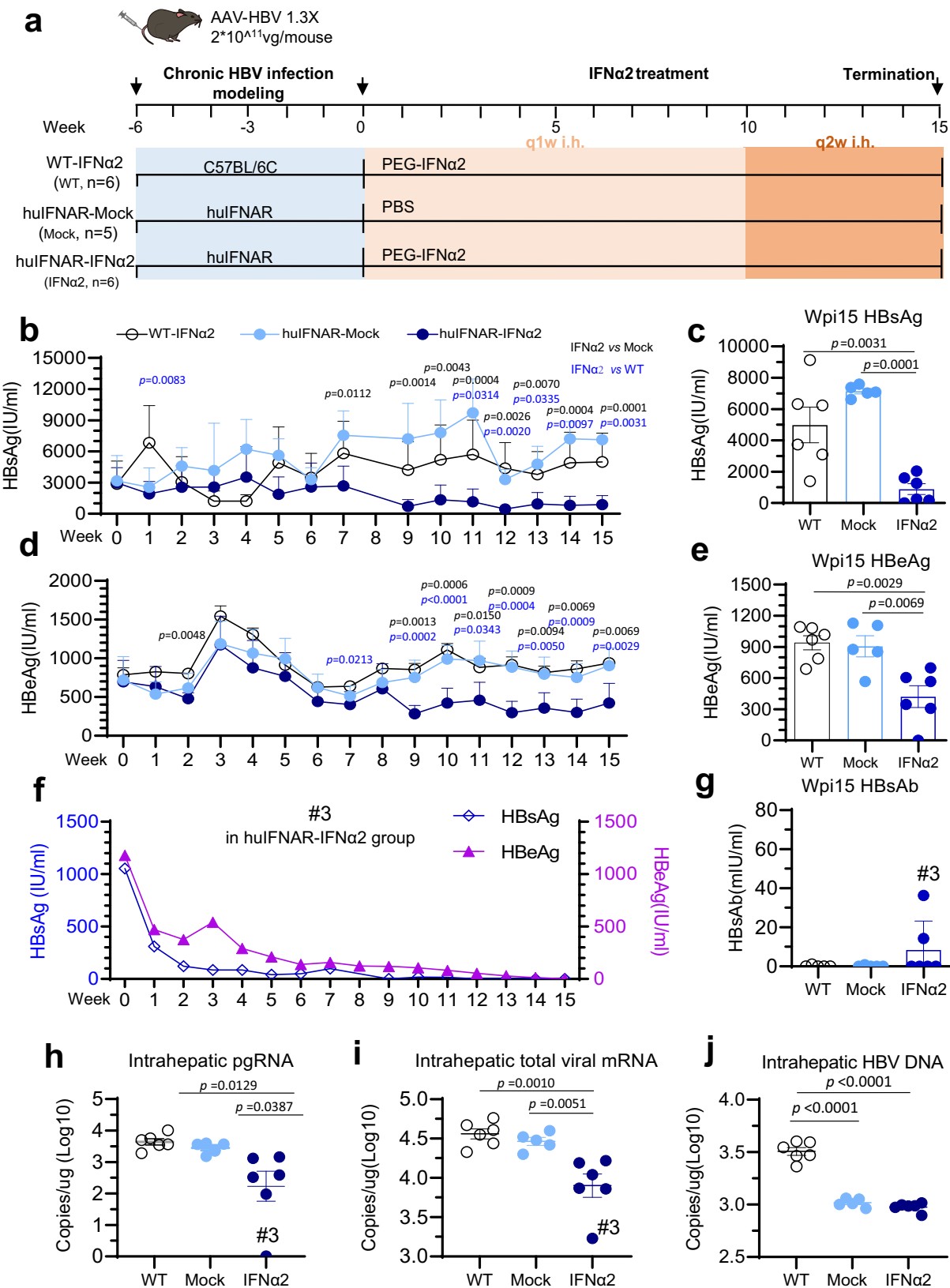

derived liver was reported to allow HBV replication. Unfortunately, human interferon failed to clear HBsAg in humanized mice[28–30], thus emphasizing the essential role of the immune system in HBsAg suppression. Instead of costly humanized mice with both human liver and immune system, generating a mouse with human interferon receptors seems attractive and affordable. In an early attempt, a human IFNAR

overexpression transgenic mouse model suffered two major defects[31]. First, the insertion of the human IFNAR gene in a conserved genomic region and under the control of a mouse phosphoglycerate kinase 1 (PGK1) promoter (non-relevant to interferon) resulted in universal human IFNAR expression in all tissues. Therefore, exogenous human IFNα stimulation inevitably elicits an artificial and non-representative

**Fig. 4 | Human PEG-IFNα2 suppresses HBV in the huIFNAR AAV-HBV mouse model. a** Experimental protocol. A huIFNAR mouse model of chronic AAV-HBV infection was established by tail vein injection of a 1.3-fold HBV genome (AAV-1.3×HBV). After that, Peg-IFNα2(2 μg/ mouse) was injected and blood was collected from tail vein for virologic index test every week. According to the type of mice and different treatments, the mice were divided into three groups: C57BL/6J AAV-HBV PEG-IFNα2-treated wildtype group (WT-IFNα2, *n* = 6), huIFNAR AAV-HBV mock-treated group (huIFNAR-Mock, *n* = 5) and PEG-IFNα2-treated group (huIFNAR-IFNα2, *n* = 6). In this figure, n = 5 or 6 biologically independent mice. (**b**) Kinetics of serum HBsAg (IU/ml). **c** Serum HBsAg at termination. **d** Kinetics of serum HBeAg (IU/ml). **e** Serum HBeAg measurement at termination. **f** HBsAg and HBeAg in the #3

huIFNAR mouse treated by PEG-IFNα2. **g** Serum of HBsAb levels. Intrahepatic HBV biomarkers at termination. Intrahepatic HBV pgRNA (**h**), HBV total mRNA (**i**) and HBV DNA (**j**) levels were measured by qPCR. Two-sided ANOVA test or Kruskal–Wallis test was performed in (**b–e**) and (**h–j**). The p values in black represent comparisons between the huIFNAR-IFNα2 group and huIFNAR-Mock group, while the ones in blue are between the huIFNAR-IFNα2 group and WT-IFNα2 group, both adjusted for multiple comparisons. Data in (**b–e**) and (**g–j**) are presented as mean values ± SEM and their replicates same to (**a**). In **c**, **e**, and **g–j**, WT, Mock and IFNα2 represent WT-IFNα2, huIFNAR-Mock and huIFNAR-IFNα2, respectively. Source data are provided as a Source Data file.

immune response in vivo. Second, the co-existence of human and mouse IFNAR caused a dysregulated immune response. Instead, our direct disruption of the mouse R2 receptor abolishes native mouse IFNAR (Fig. 1a), rendering the native mouse type I IFN signaling deficient. Importantly, knocking in human IFNAR under the native mouse Ifnar2 promoter will maximally reflect a bona fide gene expression in vivo. Indeed, human IFNα2 elicited a human-like transcriptome profile in the huIFNAR mice (Fig. 2). Therefore, this huIFNAR mouse provides a convenient preclinical model to evaluate IFN function in vivo in multiple applications.

Comprehensively delineating the transcriptome atlas of multiple organs in response to human IFN stimulation is fascinating but impossible in healthy humans. Here, we revealed that the blood, liver and spleen exhibited the most dramatic transcriptional response upon IFN stimulation (Fig. 3). The magnitude of gene expression changes seemed to be positively correlated with the human IFNAR receptor levels except in the heart, with a markedly shrunk profile. The current atlas analysis is still preliminary. Longitudinal observation with advanced technologies (such as single-cell sequencing, spatial transcriptomics etc.) under native and various infection conditions would be more informative.

In the huIFNAR mouse, human PEG-IFNα successfully reduced HBsAg after a 15-week treatment, while HBsAg in the wildtype mice showed no decline (Fig. 4), which further corroborates the potential value of these huIFNAR mice for HBV drug development. In our huIFNAR mice, HBsAg suppression by human PEG-IFNα2 took a relatively long time, resembling our observation in clinical practice that HBsAg loss often takes several months after IFNα treatment in CHB patients[8,12–14,32]. However, in wildtype AAV-HBV C57BL/6J mice, a 2-week mouse IFN treatment could easily inhibit HBsAg[19], which is far different from real observations in CHB patients. Due to the lack of a human immune system in liver-humanized mice, serum HBV-DNA and HBsAg levels continued to increase even after a 2-week daily human IFNα2 treatment[30], underscoring the vital role of immune cells in controlling HBV. Interestingly, one mouse achieved HBeAg loss and HBsAg seroconversion in our experiment. Prolonged IFNα treatment might result in HBsAg loss in more mice. Of note, human IFNα2 has a less potent antiviral effect than other members of the type I interferon family[20,29,30]. Thus, the current model will be valuable for testing the antiviral potencies of various human IFNs in vivo.

The co-expression of exhaustion-related biomarkers in intrahepatic CD8+ T effector cells provides an insight into HBV treatment. Eliminating the inhibition of Pdcd1, LAG3, CTLA4 and HAVCR2 might activate the cytotoxicity of effector T cells in suppressing HBV. Indeed, co-administration of PEG-IFNα2 and anti-PD-1-based immunotherapy resulted in significantly enhanced anti-tumor effects in two clinical trials[33,34]. The combination of PD-1 blockade with Peg-IFNα could restore CD8+ T-cell cytotoxic capacity and exert a significant synergistic effect on HCC. A recent paper by Hua Peng's lab demonstrated that targeting the liver with an engineered anti-PDL1-IFNα heterodimer can break HBV-induced immune tolerance to an HBsAg vaccine[16]. Our observation suggests that further inclusion of additional immune

checkpoint blockages might improve the effect of PEG-IFNα2, offering a promising strategy for the functional cure of CHB.

Our study has some limitations. First, we observed no HBV-DNA decline after IFNα2 treatment. Under native HBV infection conditions, all viral mRNAs might be synchronously regulated due to their overlapping genomic organization. The artificially linearized HBV 1.3-fold genome employed in the AAV-1.3 × HBV model might break this synchronization, even though an accurate representation of HBV DNA and proteins is produced. We will confirm our speculation in a mouse model with recombined HBV cccDNA. Second, the huIFNAR mice produced approximately 100-fold less serum HBV viral DNA than the wildtype mice, despite equal levels of serum HBsAg and HBeAg after tail vein injection of the same number of AAV-1.3 × HBV particles. We still don't know why this happened. We guess that some unknown factors in the type I interferon-regulated network contribute to the suppression of HBV rcDNA generation. The underlying mechanism warrants further investigation. Third, the still kept mouse *Ifnar1* in the current huIFNAR mouse might bring side-effect if used to test mouse IFN in this model. We are generating a new version of the huIFNAR mouse without the mouse *Ifnar1* gene.

In conclusion, we successfully generated a huIFNAR mouse model that can authentically mimic the human IFN response in vivo. This model revealed tissue-specific activation in response to human IFNα stimulation, which is impossible to accomplish in healthy humans. Long-term human IFNα treatment achieved HBsAg decline and loss, highlighting the potential application of huIFNA mice in human IFN-related antiviral and vaccine development and evaluation.

## Methods
### Ethical compliance
All animal experiments were conducted following Chinese guidelines for housing and care of laboratory animals and per protocols approved by the Institutional Animal Care and Use Committee (No. 2016-153) within the Guangdong Province Academy of Agricultural Sciences Animal Hygiene Institute. And all mouse experiments followed the guidelines developed by the National Centre for the Replacement, Refinement and Reduction of Animals in Research (NC3Rs).

Human specimen research was approved by Guangzhou Eighth People's Hospital Ethics Committee (No. 202001134 and 202115202). Written informed consent was obtained from volunteers.

### Mouse stocks, maintenance
C57BL/6J mice were purchased from GemPharmatech Co., Ltd (Nanjing, China). Mice were maintained on-site in the Guangzhou XY Biotechnology Co., Ltd.

### Plasmid construction
A code-optimized huIFNAR sequence, as indicated in Fig. 1a, was synthesized by GenScript Biotechology Corporation (Nanjing, China). The donor plasmid was generated through several rounds of overlapping PCR to include the left and right recombinant arms and the polyA signal sequence. The plasmid sequence information is provided in Supplementary Data 1.

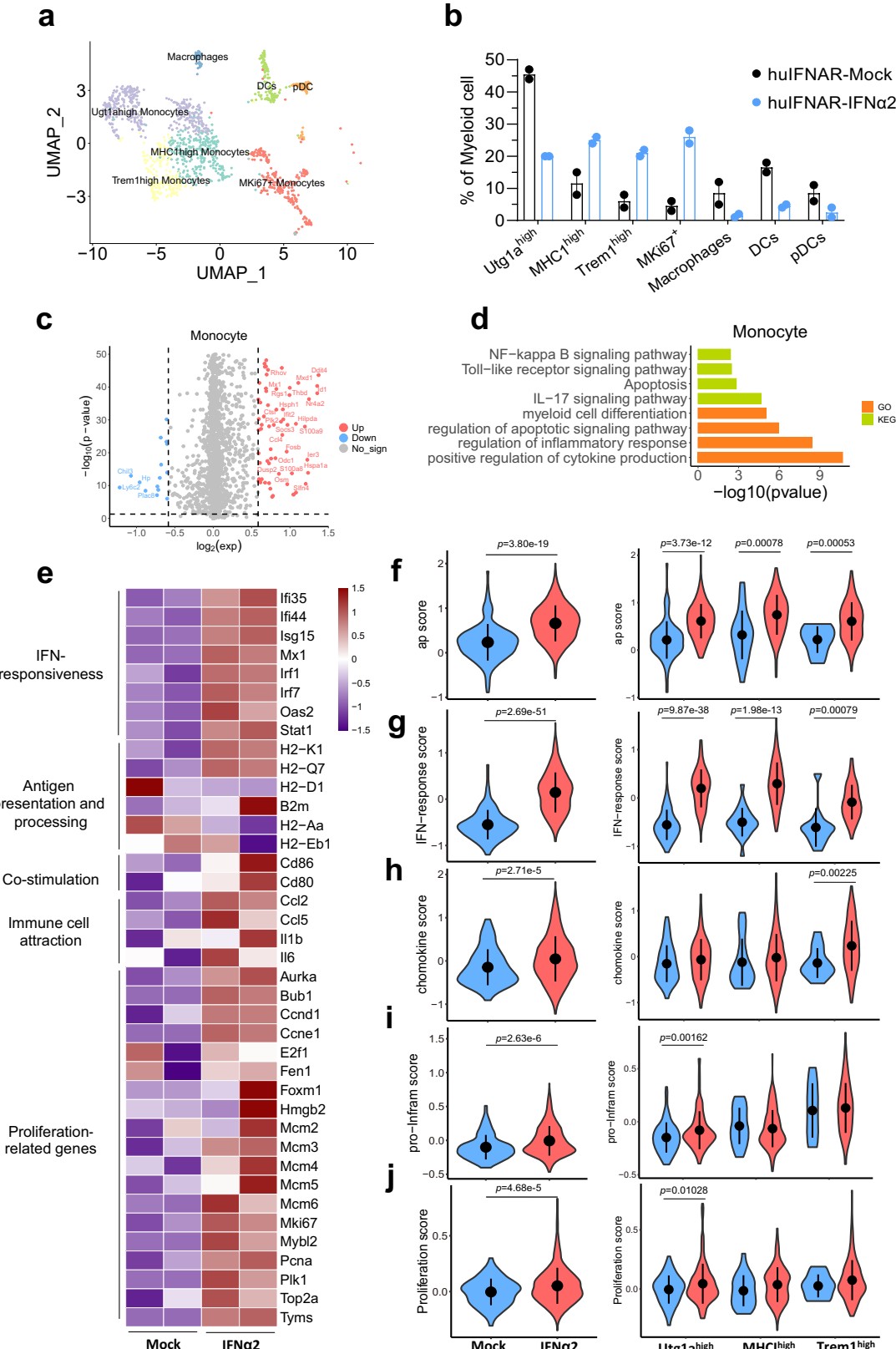

**Nature Communications** | (2023)14:7393

## Humanized type I IFN receptor knock-in mouse (huIFNAR)

The huIFNAR mice were established as illustrated in Fig. 1a. Female C57BL/6J mice (4–10 weeks of age) were super-ovulated by intraperitoneal injection with 5 IU Pregnant Mare Serum Gonadotropin (PMSG), followed by a 5 IU Human Chorionic Gonadotropin (HCG) at 48 h later. Experienced male C57BL/6J mice (5 months of age) were mated with the superovulation female at 16 h later. Plugged females were sacrificed 14–16 h following mating. Oviducts were collected, and oocyte-cumulus complexes were released from the oviducts. Fertilized embryos with visible pronuclei were selected for pronuclear microinjection and transferred to microinjection dishes containing M2 medium under mineral oil. The CRISPR/Cas9, sgRNA and linearized

**Fig. 5 | Characteristics of the intrahepatic monocytes post a 15-week interferon treatment at single-cell level. a** The uniform manifolds approximation projection (UMAP) of myeloid cell clusters. Eight populations visualized from 941 myeloid cells were indicated with different colors. **b** Population constitution analysis between MOCK (black) and IFNα2 (blue) treated groups, $n = 2$ cell samples examined independently, which were obtained and pooled from 2 or 3 mice. Data are presented as mean values ± SEM. **c** Volcano plots of differentially expressed genes in monocytes between MOCK and IFNα2 mice. Negative log2 fold change indicates downregulation (blue), positive log2 fold change indicates upregulation in IFNα2 (red) relative to MOCK mice. Genes with a log2 fold change between −0.5 and 0.5

are shown in gray. **d** Kyoto Encyclopedia of Genes and Genomes (KEGG) and gene ontology (GO) analysis of differentially expressed genes in IFNα2 versus MOCK livers. Top 4 significantly altered pathways are presented. **e** Heatmap of the enriched genes in the IFNα2-treated group. Biological processes are indicated. **f–j** Violin plots of gene set enrichment analysis scores between the huIFNAR-Mock and huIFNAR-IFNα2 groups. Data were analyzed using two-sided *t*-test. Only *p* values less than 0.05 are indicated, and *n* = total number of cells in each group. In **e–j**, WT, Mock and IFNα2 represent WT-IFNα2, huIFNAR-Mock and huIFNAR-IFNα2, respectively. Source data are provided as a Source Data file.

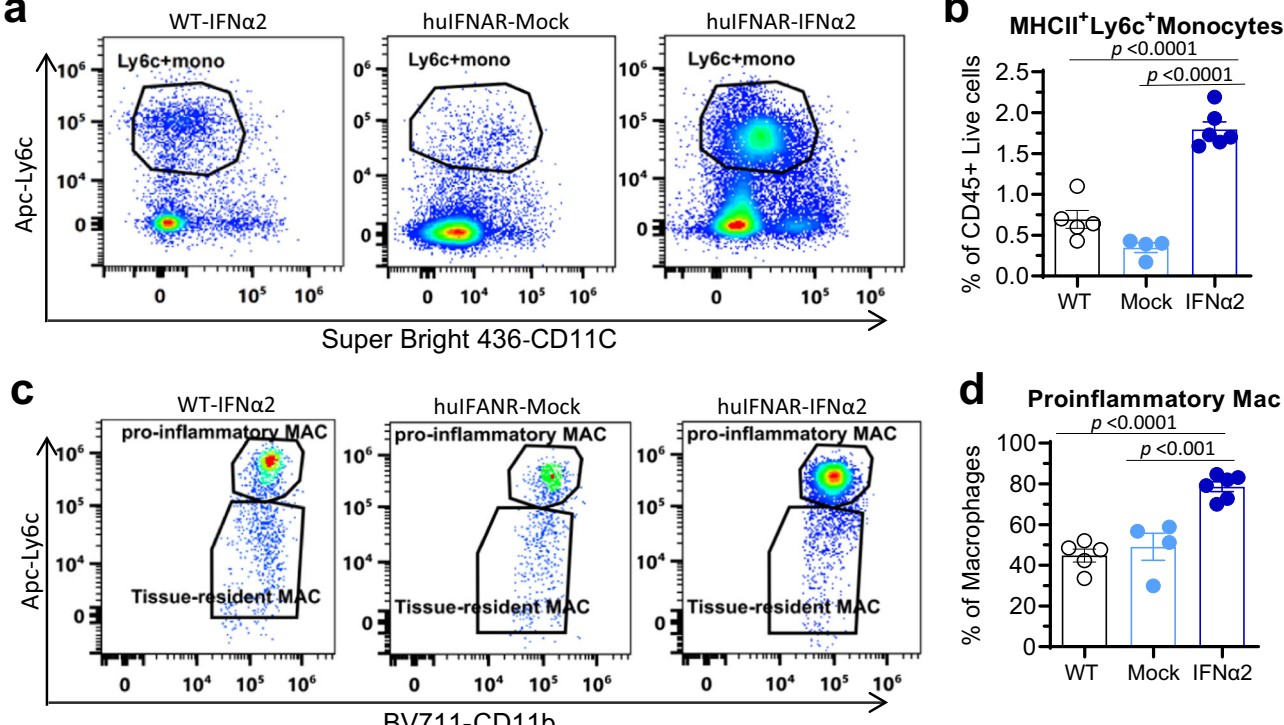

**Fig. 6 | Flow cytometry analysis of the intrahepatic monocyte and macrophages. a, c** Representative gating of specific immune cells population of MHCII⁺ Ly6c⁺Monocytes, proinflammatory (MHCII⁺ Ly6cʰⁱᵍʰ) Macrophages, respectively. **b, d** The cell frequencies among C57BL/6J AAV-HBV PEG-IFNα2-treated wildtype group (WT-IFNα2, $n = 6$), huIFNAR AAV-HBV mock-treated group (huIFNAR-Mock,

$n = 5$) and PEG-IFNα2-treated group (huIFNAR-IFNα2, $n = 6$). Data were analyzed using ANOVA test or Kruskal–Wallis test as appropriate. Data are presented as mean values ± SEM. In **b, d**, WT, Mock and IFNα2 represent WT-IFNα2, huIFNAR-Mock and huIFNAR-IFNα2, respectively. Source data are provided as a Source Data file.

huIFNAR DNA mixture were prepared and introduced into the pronuclei of fertilized embryos by microinjection using a continuous flow injection mode. Surviving embryos were surgically implanted into the oviducts of pseudo-pregnant Swiss Webster recipient females[35]. The experimental protocols were approved by Institutional Animal Care and Use Committee (No. 2016-153). The experiments were conducted in accordance with the Guidelines for the Care and Use of Laboratory Animals of Guangdong Province Academy of Agricultural Sciences Animal Hygiene Institute (Guangzhou, China).

## RNA extraction and quantification of RNA levels
Total RNA was extracted from tissues using RNeasy with DNase treatment (Qiagen). The extracted RNA was reverse transcribed using First Strand cDNA Synthesis KitReverTraAce -α- (TOYOBO, Waltham, MA, USA). Quantitative PCR was carried out using Thunderbird SYBR qPCR Mix (Toyobo) and the StepOnePlus™ Real-Time PCR System (Bio-Rad CFX96 Deep well). Mouse GAPDH and Actin mRNAs were used as the housekeeping genes. Primers were listed in Supplementary Table 1.

## Next-generation RNA sequencing and analysis
Total RNA was isolated and then purified by mRNA Capture Beads (VAHTS, N401-02). RNA library construction was performed following the manufacturer's instruction (MGIEasy RNA Library Prep Kit, 1000006383, and VAHTS Universal V6 RNA-seq Library Prep Kit for lllumina, NR604-01). The qualified RNA library was sequenced with the MGISEQ-2000 platform (MGI, Shenzhen, China) in a 100-double-end sequencing method or the NovaSeg 6000 platform (Illumina, San Diego, USA) in a PE150 sequencing method. Raw reads were trimmed with trim_galore and cutadapt to remove low-quality reads and adapter sequences, then mapped to mouse mm10 genome or human hg38 genome downloaded from the USCS Genome Bioinformatics using STAR and HISAT2[36]. Quantification of gene expression was performed using featureCounts[37] and then filtered to remove those genes with extremely low expression. Differential expression analysis and counts per million (CPM) were obtained with "TMM" method in edgeR[38]. Genes were considered as differentially expressed if the adjusted $p < 0.05$. Gene ontology analysis was conducted with clusterProfiler[39], and KEGG with pathview[40].

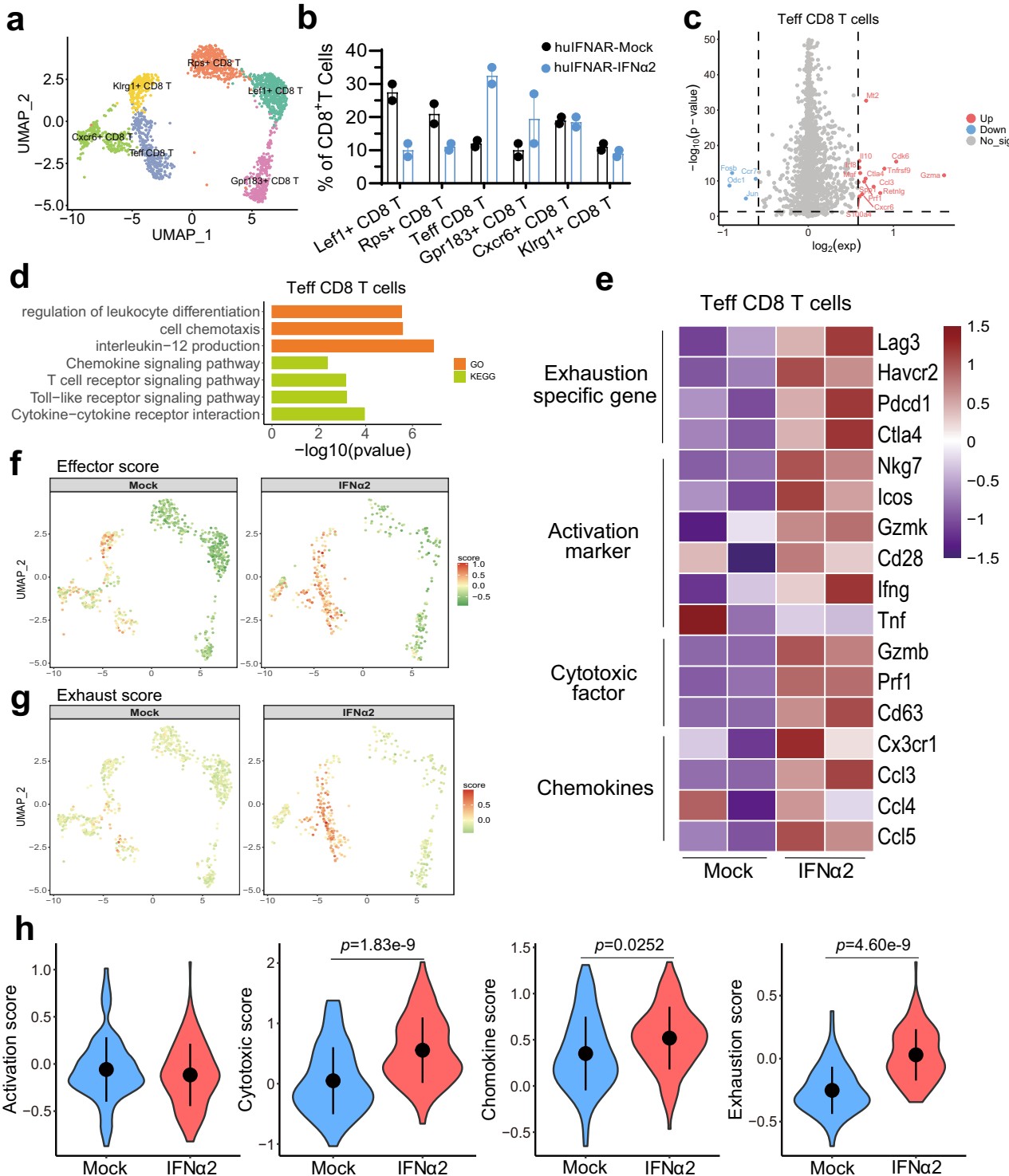

**Fig. 7 | Characteristics of the intrahepatic CD8+ T Cells post a 15-week interferon treatment at single-cell level. a** The uniform manifolds approximation projection (UMAP) of CD8+ T-cell clusters. Six populations designated by specific gene expression visualized from 1045 intrahepatic CD8+ T cells were indicated with different colors. **b** Population constitution analysis between huIFNAR-Mock (black) and huIFNAR-IFNα2 (blue) treated groups, *n* = 2 cell samples examined independently, which were obtained and pooled from 2 or 3 mice. Data are presented as mean values ± SEM. **c** Volcano plots of differentially expressed genes in effector CD8+ T cells between huIFNAR-Mock and huIFNAR-IFNα2 mice. Negative log2 fold change indicates downregulation (blue), positive log2 fold change indicates upregulation in IFNα2 (red) relative to Mock mice. Genes with a log2 fold change between −0.5 and 0.5 are shown in gray. **d** GO and KEGG Biological Process analysis of differentially expressed genes in huIFNAR-IFNα2 group. Top 3–4 significantly altered pathways are presented. **e** Heatmap of the enriched genes in the huIFNAR-IFNα2-treated group. Biological processes are indicated. **f, g** Individual cell effector score and exhaust score overlay for selected differential canonical pathway activities (effect and depletion phenotype) in huIFNAR-Mock and huIFNAR-IFNα2 groups. **h** Violin plots of gene set enrichment analysis scores. Data were analyzed using two-sided *t*-test. Only p values less than 0.05 are indicated, and *n* = total number of cells in each group. In **e–h**, WT, Mock and IFNα2 represent WT-IFNα2, huIFNAR-Mock and huIFNAR-IFNα2, respectively. Source data are provided as a Source Data file.

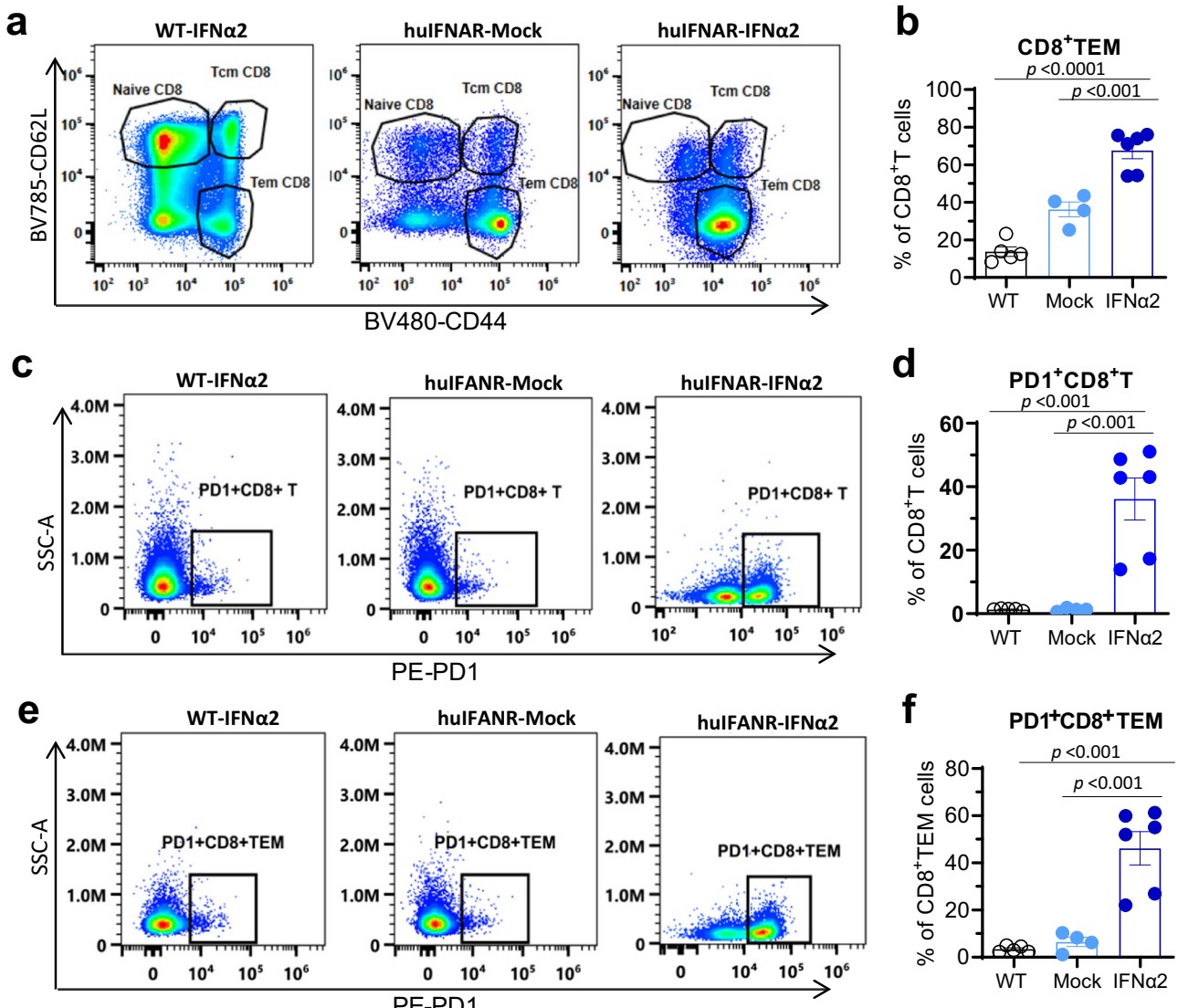

**Fig. 8 | Flow cytometry analysis of the intrahepatic CD8⁺ T cells.**
**a**, **c**, **e** Representative gating of specific immune cells population of effector memory CD8⁺ T (TEM), PD1⁺ CD8⁺T and PD1⁺CD8⁺TEM Cells, respectively. **b**, **d**, **f** The cell frequencies among C57BL/6J AAV-HBV PEG-IFNα2-treated wildtype group (WT-IFNα2, $n = 6$), huIFNAR AAV-HBV mock-treated group (huIFNAR-Mock, $n = 5$) and PEG-IFNα2-treated group (huIFNAR-IFNα2, $n = 6$). Data were analyzed using ANOVA test or Kruskal–Wallis test as appropriate. Data are presented as mean values ± SEM. In **b**, **d**, **f**, WT, Mock and IFNα2 represent WT-IFNα2, huIFNAR-Mock and huIFNAR-IFNα2, respectively. Source data are provided as a Source Data file.

## Modeling HBV infection AAV-HBV mouse and Peg-IFNα2 treatment

Male C57BL/6J mice (6–10 weeks of age, $n = 18$) were injected intravenously by tail vein with $2.0 \times 10^{11}$ viral genomes (vg) rAAV8-1.3HBV, genotype C, adr (PackGene Biotech, Guangzhou, China) in 100 µl PBS buffer. After 6 weeks, HBV-carrier mice with serum HBsAg levels > 500 IU/mL were randomized into different treatment groups based on serum HBsAg levels ($n = 5$–6/group). As shown in Fig. 4a, mice were then injected subcutaneously with 2 µg Peg-IFNα2 (Roche Pharma, Schweiz) or PBS for a 15-week treatment. Mice were euthanized within a week of the end of treatment.

## Serum HBV biomarker detection

HBV-DNA levels in serum were quantified by real-time PCR (Hepatitis B Viral DNA Quantitative Fluorescence Diagnostic Kit, Sansure Biotech, Changsha, China). The serum HBsAg level was determined using a commercial assay kit (Shenzhen Mindray Bio-Medical Electronics Company, Shenzhen, China) by chemiluminescent immunoassay

(CLIA). Serum HBeAg and serum HBsAb levels were quantified by an ELISA kit (Wantai Biological Pharmacy, China). Serum HBeAg standards (6.25, 3.125, 1.56, 0.78, 0.39, 0.195, 0.097, 0.049 IU/ml) were prepared from the stocked standard solution by using diluents of 5% BSA. Serum HBsAb standards (169.58, 84.79, 42.39, 21.20, 10.60, 5.30, 2.65, 1.32 mIU/ml) were prepared from the stocked standard solution by using diluents of 5%BSA. The amount of HBeAg and HBsAb in each sample was calculated based on the standard curve. Samples were re-tested by further dilution if the OD values were out of the range of the standard curve.

## Intrahepatic viral biomarker detection

Intrahepatic nucleic acids were isolated using TIANamp Genomic DNA kit (cat: DP304, TIAGEN Biotech, Beijing, China) and the RNeasy RNA purification kit (Qiagen). Total HBV-DNA amounts were normalized for cellular DNA contents using the human beta-globin gene kit (Roche Applied Science). Total mRNAs were reverse transcribed using oligo-dT primers. The Transcriptor Kit (TOYOBO SYBR Premix Ex Taq 2X)

was used for qPCR analysis. Primers were listed in Supplementary Table 1.

### Intrahepatic lymphocyte single-cell isolation

Intrahepatic lymphocyte cell isolation was done by enzymatic digestion[41,42]. Briefly, liver was minced into 0.5–3 mm pieces in digestion buffer (RMPI 1640 + 5%FBS + 20 ng/ml DNase I + 0.5 mg/mL Collagenase D), and enzymatically digested with the program 37C_m_LIDK_1 by gentleMACS (Miltenyi Biotec, Germany). Liver homogenates were passed through a 70-μm cell strainer (BD, USA) and then centrifuged at 300 g for 5 min. Resuspend the pellet in 5 ml of 40% Percoll and centrifuged at 435 × g for 20 min without brake. After the supernatant was removed, the pelleted cells were suspended in RBS lysis buffer and incubated on ice for 5 min to lyse red blood cells. After washing twice with PBS, the cell pellets were re-suspended in PBS (containing 0.04%BSA)[43].

### 20-color flow cytometry analysis

A 20-color multi-parametric flow cytometry was used for the phenotypic analysis of intrahepatic lymphocyte cells. Briefly, resuspend immune cells at $2 \times 10^6$–$4 \times 10^6$ cells/ml in 1 ml of DPBS (BI, 02-023-1A) and stained with 0.3 μl of FVS575V for 20 min at room temperature in the dark for live/dead. Cells were washed twice (400 × g, 5 min, 4 °C) with FACS buffer (2%FBS/PBS). The cell pellet was re-suspended in 90ul of FACS buffer. For Fc receptors blocking, add 4ul of anti-CD16/32 antibody and incubate for 15 min on ice. Stain the cells with CXCR5-BV650 for 10 min at RT in the dark and then incubated the cells with the other 19 kinds of surface receptor staining mix for another 30 min at RT in the dark. After incubation, cells were washed with FACS buffer twice and re-suspended in 200 μl FACS buffer. The flow cytometric data were collected on a spectral flow cytometry mechine (Cytek NL-CLC, Cytek Biosciences, USA) and analyzed using FlowJo software, V10.7.1 (Tree Star, USA). Detailed information about antibodies is shown in Supplementary Table 2.

### Single-cell RNA library construction and sequencing

DNBelab C Series High-throughput Single-Cell RNA Library (MGI, #940-000047-00) was utilized for scRNA-seq library preparation. In brief, the single-cell suspensions were converted to barcoded scRNA-seq libraries through steps including droplet encapsulation, emulsion breakage, mRNA captured beads collection, reverse transcription, cDNA amplification and purification. cDNA production was sheared to short fragments with 250–400 bp, and indexed sequencing libraries were constructed according to the manufacturer's protocol. Qualification was performed using Qubit ssDNA Assay Kit (Thermo Fisher Scientific) and Agilent Bioanalyzer 2100. All libraries were further sequenced by the MGISEQ-2000 or DNBSEQ-T7 sequencing platform with pair-end sequencing. The sequencing reads contained 30-bp read 1 (including the 10-bp cell barcode 1, 10-bp cell barcode 2 and 10-bp unique molecular identifiers (UMI)), 100-bp read 2 for gene sequences and 10-bp barcodes read for sample index.

### Single-cell RNA-sequencing data processing (alignment, barcode assignment, and UMI counting)

The sequencing data were processed using an open-source pipeline (https://github.com/MGI-tech-bioinformatics/DNBelab_C_Series_HT_scRNA-analysis-software). Briefly, all samples were performed sample de-multiplexing, barcode processing, and single-cell 3′ unique molecular identifier (UMI) counting with default parameters. Processed reads were then aligned to GRCh38 genome reference using STAR (v2.5.3). Valid cells were automatically identified based on the UMI number distribution of each cell by using the "barcode Ranks()" function of the Droplet Utils tool to remove background beads and the beads that had UMI counts less than the threshold value. Finally, we

used PISA to calculate the gene expression of cells and create a gene × cell matrix for each library.

The DNBelab C Series (v 2.0) provided by MGI was applied to align reads and generate the gene-cell unique molecular identifier (UMI) matrix against the reference genome mm10 downloaded from NCBI. For each cell, we quantified the number of genes and UMIs and kept high-quality cells with 200–5000 genes detected and no more than 10% of mitochondrial gene counts.

### Unsupervised dimension reduction and clustering analysis

The filtered unique molecule identifiers of genes were normalized using the NormalizeData program of the R Seurat package (v4.0.3) with default parameters. Next, the IntegrateData function was applied to correct the batch effect between MOCK and IFNα2 groups. The RunPCA program was performed based on the top 2000 highly variable genes generated by the FindVariableFeatures function, and then the UMAP of single cells was generated by the RunUMAP program. Finally, we adopted FindNeighbors and FindClusters to cluster cells into subclusters at a resolution of 0.8 and visualized them by UMAP with default settings.

T cells, including CD8 T cells and myeloid cells, were re-clustered following the similar steps described above, incorporating integration, dimension reduction, and clustering analysis.

### Identification of marker genes and annotation of cell clusters

Marker genes for each cluster were identified with the MAST algorithm in the FindAllMarkers function of Seurat. The following criteria were used to filter the marker: |log2FC| ≥ 0.583, p.adjust ≤ 0.05, and pct.1 ≥ 0.25 or pct.1 ≥ 0.25. The cell clusters were annotated by previously reported cell type-specific marker genes.

### Calculating gene expression signature scores

The cell gene expression signature score was calculated using the AddModuleScore function in Seurat. The exhaustion signature scores of the CD8 T cells were calculated using genes as follows: *Lag3, Havcr2, Pdcd1, Ctla4, Entpd1, Tigit, Tnfrsf9, Cd27, Layn, Eomes, Tox, Tox2, Hopx, Arnt, Etv1* and *Irf8*. The activation signature scores of the CD8 T cells were calculated using genes as follows: *Icos, Fcgr3a, Fgfbp2, Cd28, Etv7, Prdm1, Blimp-1, Stat3, Egr1* and *Egr2*. The cell function scores of the CD8 T cells were calculated using genes as follow: *Ifng, Tnfa, Il2*. The cytotoxicity signature scores of the CD8 T cells were calculated using genes as follows: *Gzmb, Prf1, Gnly, Cd63, Nkg7, Gzmk, Gzma* and *Gzmh*. The chemokine scores of the CD8 T cells were calculated using genes as follows: *Cx3cr1, Ccl3, Ccl4, Ccl5, Cxcl10, Cxcr3* and *Cxcr4*. The effector scores of the CD8 T cells were calculated using genes including *Lag3, Havcr2, Pdcd1, Ctla4, Entpd1, Tigit, Tnfrsf9, Cd27, Layn, Eomes, Tox, Tox2, Hopx, Arnt, Etv1, Irf8, Icos, Fcgr3a, Fgfbp2, Cd28, Etv7, Prdm1, Blimp-1, Stat3, Egr1, Egr2, Gzmb, Prf1, Gnly, Cd63, Nkg7, Gzmk, Gzma* and *Gzmh*. The antigen presentation signature scores of the monocyte cells were calculated using genes as follows: *Cd80, Cd86, H2-K1* and *H2-Q7*. The chemokine signature scores of the monocyte cells were calculated using genes as follows: *Ccl2, Ccl5, Ccl6, Cxcl9, Cxcl10* and *Cxcl2*. The proinflammatory signature scores of the monocyte cells were calculated using genes as follows: *Ifng, Il1r1, Tlr4, Il6, Il1b, Tnf, Csf2, Tlr2, IL1a, Nos2, Ly6c1, Marco* and *Nlrp3*. The interferon response signature scores of the monocyte cells were calculated using genes as follows: *Ifng, Il1r1, Tlr4, Il6, Il1b, Tnf, Csf2, Tlr2, IL1a, Nos2, Ly6c1, Marco* and *Nlrp3*.

### Functional annotation analysis

The FindMarkers function was applied to detect the DEGs from a pairwise comparison. The following criteria were used to define DEGs: |log2FC| ≥ 0.583, p-value ≤ 0.05, and pct.1 ≥ 0.25 or pct.2 ≥ 0.25. GO and Kyoto Encyclopedia of Genes and Genomes (KEGG) pathway analysis of DEGs were performed with clusterProfiler R package. Only terms in the GO Biological Processes were considered in the GO enrichment

analysis. In addition, GSEA was also included and performed with C5 (Gene Ontology) in MSigDB.

### Statistical analysis

Continuous variables were expressed as median (interquartile range, IQR). Categorical variables were summarized as the counts and percentages in each category. Unpaired *t*-test, One-way ANOVA tests or Kruskal–Wallis tests were applied to continuous variables as appropriate; chi-square test or Fisher's exact test was applied to categorical variables as appropriate, log-rank (Mantel–Cox) test was applied to virus RNA clearance, $p < 0.05$ was considered statistically significant. Statistical analysis and graphic representations were performed with GraphPad Prism 8.0.1 software. RNA-sequencing data processing and analysis were performed with fastp, STAR, featureCounts, edgeR (v.3.32.1) packages, fgsea (v.1.16.0) packages of R studio. The flow cytometric data were analyzed using FlowJo software V10.7.1.

### Reporting summary

Further information on research design is available in the Nature Portfolio Reporting Summary linked to this article.

## Data availability

All tissue-specific bulk RNA-seq and single-cell RNA-seq data integral to this study have been responsibly deposited in the National Center for Biotechnology Information Gene Expression Omnibus, accessible via the accession codes GSE237519. Complementary bulk RNA-seq data are made available in the Genome Sequence Archive (GSA) or GSA-Human, under the BioProject accession PRJCA017918. The raw data related to individual sequences in PRJCA017918 are available under restricted access for data privacy laws; access can be obtained by submitting a formal request to the corresponding author, following the guidelines provided on the website. Additional datasets used in the study include the mouse mm10 genome [https://www.ncbi.nlm.nih.gov/datasets/genome/GCF_000001635.20/], human hg38 genome [https://www.ncbi.nlm.nih.gov/datasets/genome/GCF_000001405.26/], and KEGG pathways [https://www.genome.jp/kegg/pathway.html]. The remaining data that support our findings are available within the article, Supplementary Information, or Source Data file. Source data are provided with this paper.

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

## Acknowledgements

We thank colleagues in the mouse facility for take care the mouse, and Zichang Wang, Yajun Zuo, Zhe Zhang, Xuan Gong, Ming Ni and Longqi Liu in MGI Tech Co., Ltd (Shenzhen, China) for technical assistance in sequencing. This work was supported by the National Natural Science Foundation of China (Nos. 81670536 and 92269108 to F.L., 81770593 to X.T.), the National Grand Program on Key Infectious Disease Control (Nos. 2017ZX10202203-004-002 and 2018ZX10301404-003-002 to F.L.), the Guangzhou Science and Technology Planning Project (Nos. 202102080389, 202206080001 and 2022-01-02-04-0087 to F.L.; 2023A03J0804 to Y. Wang; 202201020251 to C.F.), Key-Area Research and Development Program of Guangdong Province (2022B1111020002 to X.T.) and Guangdong Provincial Natural Science Foundation (2021A1515011287 to F.L.). Guangdong Basic and Applied Basic Research Foundation (No. 81670536 to Y. Wang).

## Author contributions

F.L., X.T., L.Z. and Y. Wang designed and supervised the experiments, analyzed and interpreted the data. F.L. and Y. Wang wrote the manuscript. Y. Wang, L.G., J.S., J.L. performed the cellular and most mouse experiments and prepared figures. Y. Wen, Y. Wang and Z.Z. analysis the single-cell sequencing data. G.G., C.F., X.Z. generated the AAV-HBV mouse model. J.C., S.C., J.T. and J.Z. measured HBV DNA, HBsAg, HBeAg and HBsAb. M.J., Q.F. and J.S. perform RNA sequencing and single-cell sequencing. J.S. and X.N.L. perform RNA-seq analysis. Y.S. helped the 20-color FACs analysis. G.G. and X.L. managed the mouse experiment. M.P. did the western blot analysis. F.H. critically reviewed the study.

## Competing interests

The authors declare no competing interests.
