## [Peer Review File · Nature Communications]

Interferon stimulated immune profile changes in a humanized mouse model of HBV infectionEditorial Note: Parts of this Peer Review File have been redacted as indicated to remove third-party material where no permission to publish could be obtained.

REVIEWER COMMENTS

Reviewer #1 (Remarks to the Author):

The manuscript by Wang and others described a novel transgenic mouse model for the treatment with IFNs during chronic HBV infection. The mice have been analyzed in great detail, and infection studies with AAV-HBV revealed an antiviral effect of Peg-IFN α 2 with complete loss of HBsAg and HBeAg in one single mouse. This mouse model is suitable for analysis of human IFNs as these mice showed authentic responses to human IFNs. The study is well conducted and well written. However, some concerns remain which should be addressed:

1. The authors nicely explained in the discussion that the newly generated transgenic mouse strain only expresses the human IFNAR1 and 2 and not the extracellular part of the mouse IFNAR1/2. Have the others analyzed if the cells can still respond to murine IFNs *in vitro*? During the infection their might be also a low induction of murine type I IFNs, which could bind to the receptor and induce signaling. Furthermore, treatment with Peg-IFN might also induce the expression of various IFNs by a positive feedback loop and thus affecting the IFN-mediated response. Have the other analyzed the systemic IFN expression after injection of Peg-IFNs?

2. The authors compared the effect of transgenic mouse PBMCs after *in vivo* Peg-IFN injection (10 μ g/mouse s.c.) and *in vitro/ex vivo* Peg-IFN stimulated (0.8 μ g/ml) human PBMCs. I would appreciate a direct comparison of transgenic mouse PBMCs and human PBMCs *in vitro* with the same Peg-IFN concentration. Maybe also a kinetic of early and late expressed ISGs (e.g., 6h vs. 18-24h) in *in vitro* IFN-stimulated human and mouse PBMCs might be of interest. How were the Peg-IFN concentrations selected for the different experimental conditions (*in vitro/ex vivo* vs. *in vivo*). In particular during AAV-HBV infection experiment, multiple injections of 2 μ g per mouse were performed, whereas in the initial *in vivo* experiment 10 μ g/mouse was injected.

3. The authors analyzed very precisely the expression of the human IFNAR2 in various tissues. As the authors later also focused on various immune cell subsets, they should also analyze the expression of IFNAR1 and IFNAR2 in different immune cell subsets like T cells, NK cells, neutrophils, myeloid cells, and B cells.

4. Did the Peg-IFN-treated mice develop any side effects like fever or cytokine storm? As only one mouse (#3) achieved HBsAg and HBeAg loss during treatment, different concentrations of Peg-IFN should be considered *in vivo*.

5. The data suggest that Peg-IFN treatment strongly affected CD8⁺ T cell responses during HBV infection. Apart from the extensive gene expression analysis, flow cytometry analysis of T effector cells showing cytotoxicity (GzmB, GzmA, perforin, CD107a, TRAIL, Fas, etc) or cytokine production (IL-2, TNF α , IFN γ , etc) should be added to the study. Additionally, the analysis of core/env-specific T cells by tetramer staining or *in vitro* peptide stimulation after treatment with PEG-IFN might be included to the manuscript. Furthermore, CD8 T cells during Peg-IFN treatment should be analyzed in depletion experiments to elucidate their exact role during IFN-therapy.

6. Please change the order of the figures: Fig.7 is mentioned in the text before Fig. 6 and Supp.Fig.14 and Supp. Fig.15 before Supp. Fig.13.

Reviewer #2 (Remarks to the Author):

Reviewer comments on "Comprehensive immune profile changes to human interferon stimulation in normal and HBV-infected conditions in a humanized mouse".

The authors have established a human IFNAR knock-in mouse model which they tested using *in vivo* and *ex vivo* treatment with human IFN α . Furthermore, they applied the IFN treatment in the context of the AAV-HBV model. They also performed bulk and single-cell RNA-Seq analysis and also looked into specific ISGs using qRT-PCR. Despite the extensive work that is presented here,

essential controls are missing that would highlight the importance for this mouse model and its application in IFN-related studies including treatment of HBV. Below there are specific comments figure by figure for consideration by the authors.

In Figure 1 E-H it would be more helpful to include a WB for mMx2 or mISG15. This should be quite easy since these are ex vivo samples from WT and HuIFNAR mice stimulated with human IFNa2. An essential control for this experiment is also treatment of the cells with murine IFNa. This will also allow for a comparison between the two systems but also confirm that there is no mouse-dependent stimulation in HuIFNAR. In addition, in Figure 1 I-J injection of murine IFNa would serve as a good control with both mRNA and protein levels of mMX1 and mISG15 shown. Is there a reason for switching from Mx2 in panels E and G to Mx1 in panel I? In panel 1C I would also suggest showing protein levels of HuIFNAR.

In Figure 2A/Supplementary Figure S4 the experimental design would have been better controlled if the murine PBMCs were treated with IFNa after isolation similar to what happened with the human PBMCs. Maybe it would be best to include a third group that is similar to how the stimulation was done in human PBMCs. In Figure 2B these numbers exclude any ISGs that are not found in both humans and mice? This was not clear in the text.

In Figure 3 to better appreciate the value of these data I would propose to include as a control treatment with mouse IFNa in a WT mouse. If the results are identical then what new did we learn from this experiment/model?

Line 267: FigS7 is probably FigS8. The AAV HBV 1.3X is used in HBV research but since there is no cccDNA present in the mouse hepatocytes, modeling chronic HBV infection is incomplete. This is an inherent drawback of this approach and not the manuscript's but it should be mentioned for clarification.

Figure 4 should include a control with murine IFNa on C57BL/6C.

Figure 4F should include some information on the state of the hepatocytes to exclude toxicity due to IFN treatment. The starting point for HBsAg is also 1/3 of what is shown in panel 4B with the scale being modified.

Figure 5. The single cell data would benefit from an earlier time point because at week 15 there might be increased toxicity accumulated from IFN treatment.

For some reason Fig. 7 is mentioned before Fig. 6 which is confusing.

Minor but would need to be addressed: English proofreading would make it easier to read some parts even though the majority of the manuscript is well-written.

Reviewer #3 (Remarks to the Author):

The manuscript by Wang et al. aims to analyze the mechanisms of action of human IFNa to HBV via the human IFNAR by developing a model with a knock-in of the human *Ifnar* in the mouse *Ifnar2* locus.

The data sets need to be clarified, some experimental controls are missing and the conclusions are not substantiated by the experiments with too small cohorts. Some of the statements do not reflect the literature in the field and are forgetting important concepts.

The proper use of English: spelling, grammar, and incorrect word usage must be revised throughout the manuscript.

The idea that IFNa therapy is essential to HBV functional cure is the authors' opinion and should not be stated as a fact, such sentences must be revised: page 6 line 112-114 "The IFNa-plus corroboration strategy on top of most HBV direct antivirals, subsequentially or simultaneously,

seems inevitable to achieve sustained HBV functional cure." HBV functional cure occurs more frequently without IFNa treatment than with it.

The species cross-reactivity of human to mouse IFNa needs to be clarified by in vitro and in vivo experiments, analysis of sequences as done Fig S1 and S2 is insufficient. Experiments showing the mouse IFNAR response (or lack thereof) to hIFNa2 are required to justify the importance of establishing a humanized IFNAR mouse model.

The knock-in of the human IFNAR1 and IFNAR2 genes into the mouse IFNAR2 locus must be validated and shown by sequencing.

The mouse genotyping shown in Fig S3 is confusing (and the figure legend is wrong in describing the Hetero: "Hetero, homozygous C57BL/6J-huIFNAR mouse"): why does the Homo show identical amplification with primer pairs #1 and #2 when they should be of different sizes like shown for the Hetero?

Fig 1D: what cells were used to analyze hIFNAR expression by FACS? Why are all the cells positive?

Is there residual mouse IFNAR expression? Show the knock-in is complete by FACS.

What is the residual mouse IFNAR response in the HuIFNAR mice when stimulated with mouse IFNa?

The functional assessment of hIFNa2 responses in the spleen shown in Fig 1I and 1J is problematic: mMX1 expression was present in only 2/4 mice and mISG15 0/4 mice. How do the authors explain this lack of response in the spleen? It seems unlikely that there is any statistical significance Fig 1J in the spleen ($p=0.0369$), as there is no expression! The text is not truthfully describing the results.

It is not clear why the GO category "response to type I interferon" is close to zero in Fig 2C although these cells were stimulated with PEG-IFNa2, can the authors explain why?

Fig S5A: most of the genes are illegible.

Fig 3C the gene names must be aligned with the heatmap, or else we can not analyze the data.

The AAV-1.3xHBV genotype C injected mice show a profound drop in HBsAg levels at week 2-3 which is not paralleled by the HBeAg nor the HBV DNA levels (Fig S7). This is very uncommon and has not been described by other groups. Please explain.

How do the viral kinetics in huIFNAR mice compare to wild type mice? This control is necessary.

To properly evaluate the effect of PEG-IFNa treatment in the huIFNAR AAV-1.3xHBV mice we need to see the viral kinetics prior to the PEG-IFNa treatment in Fig 4 (-6w to 0w). The HBeAg data needs to be revised to match between Fig 4 (IU/ml) and Fig S7 (OD450).

What are the HBV DNA kinetics throughout the experiment (show before IFNa treatment)?

Fig S9 shows that the groups are biased: the WT mice start with higher levels of HBV DNA than the Mock or IFNa2 treated mice. The bias in the experimental groups is also visible in Fig 4J: the WT mice have a higher level of intrahepatic HBV DNA than the Mock or IFNa groups. Why do the huIFNAR mice have lower HBV DNA levels than controls?

Seroconversion of HBs has been described in AAV-1.3xHBV mice since the mouse immune response is fully competent to develop humoral immunity (Huang YH et al. *Int J Oncol* 2011). Thus mouse #3 is not uncommon and could occur without huIFNAR if larger cohorts of animals are used.

Figure and legends are inverted for Fig S8 and Fig S9.

Text needs major revision, page 14 line 299 "As IFNs do not directly act on the intracellular HBV viral genome" this is false and forgetting some important articles: Zhang M et al. *J Hepatol* 2021, Baudi I et al. *Plos Pathogens* 2021, Allweiss L et al. *Gut* 2022.

Overinterpretation of the data from two mice in Fig S10D: no statistics are possible, the authors

can not state page 15 line 310-312 "the PEG-IFN α 2 treatment substantially increased...". This is the same issue in Fig 5B: only two mice were analyzed and the text describes these results as significant, although no statistics can be run.

Fig 7. What are the control mice? How was the liver prepared for the intrahepatic leukocyte analysis? Liver resident macrophages need to be stained with CD68. Proinflammatory macrophages are mostly in the circulating blood. Expression of PD-1 on CD8+ T cells does not necessarily mean the cells are exhausted since PD-1 is upregulated on activated cells.

Materials and Methods

- Remove the chapter "Study design" which has no reason to be in the manuscript.
- Plasmid construction details must be included, "Details will be provided upon request" is not acceptable.
- The preparation of the liver leukocytes is missing.

Point-to-point response

Reviewer #1 (Remarks to the Author):

The manuscript by Wang and others described a novel transgenic mouse model for the treatment with IFNs during chronic HBV infection. The mice have been analyzed in great detail, and infection studies with AAV-HBV revealed an antiviral effect of Peg-IFN α 2 with complete loss of HBsAg and HBeAg in one single mouse. This mouse model is suitable for analysis of human IFNs as these mice showed authentic responses to human IFNs. The study is well conducted and well written. However, some concerns remain which should be addressed:

Response: We express our heartfelt gratitude for your encouraging recognition of our study. It is evident from your insightful comments that you possess a commendable understanding of the interferon field. The complexity of human interferon family members, along with their species-specific characteristics, is often overlooked. A common assumption within the scientific community is that mice and humans share a similar interferon-stimulated downstream signaling cascade, and that all interferon family members induce identical gene expression profiles.

In our study, we comprehensively characterized PBMC gene expression profiles in response to human and mouse interferon stimulation *in vitro*, employing transcriptome sequencing. We observed remarkable changes in gene expression profiles. Importantly, our findings revealed that mice exhibit a significantly distinct gene expression profile compared to humans.

We concur with your observation that this mouse model is apt for analyzing human interferons, as these mice demonstrated authentic responses to human interferons. To the best of our knowledge, this pioneering study addresses the crucial issue of HBV functional cure by human interferon in an interferon receptor-humanized mouse model. Your insightful and constructive suggestions have been invaluable in enhancing the quality of this manuscript. We have diligently conducted all the recommended experiments.

We once again express our deep appreciation for your expert guidance and feedback.

1. The authors nicely explained in the discussion that the newly generated transgenic mouse strain only expresses the human IFNAR1 and 2 and not the extracellular part of the mouse IFNAR1/2. Have the others analyzed if the cells can still respond to murine

IFNs in vitro? During the infection there might be also a low induction of murine type I IFNs, which could bind to the receptor and induce signaling. Furthermore, treatment with Peg-IFN might also induce the expression of various IFNs by a positive feedback loop and thus affecting the IFN-mediated response. Have the other analyzed the systemic IFN expression after injection of Peg-IFNs?

Response: We appreciate your positive remarks regarding our study design.

Following your suggestion, we assessed the responsiveness of the huIFNAR mouse to mouse stimulation and selected mouse IFN α 5 due to its ready availability. Mouse PBMCs isolated from wild-type C57BL/6C and huIFNAR mice were stimulated with huIFN α 2 and mIFN α 5 for 8 h and 20 h in vitro as indicated. The ISG gene expression changes among groups were shown in the heatmap (now provided as supplementary figure 6). We observed that human IFN α 2 could not evoke obvious gene expression in mouse PBMCs compared to the mouse IFN α 5.

As for the huIFNAR mouse PBMCs, human IFN α 2 simulated a different pattern from the wildtype mouse PBMC. Meanwhile, huIFNAR mouse PBMCs also responded to mIFN α 5 stimulation. Although in a likely manner, the exaction pattern and magnitude differ substantially at both 8 h and 20 h post-stimulation. As the wildtype mouse IFNAR1 subunit remains intact in our design, we postulated that the mouse IFNAR1 and humanized IFNAR2 can form dimmers and still maintain their response to mouse IFN α stimulation.

Your valuable comments make us determined to further knockout the mouse IFNAR1 on top of the current huIFNAR mouse. We have started to generate this mouse recently. But it will take at least one year to get a colony of homozygous huIFNAR and mouse IFNAR1 knockout mice to test this postulation.

Figure 1 for review: Transcriptomic Response to Human and Mouse interferons in Wildtype and HuIFNAR Mice. The figure illustrates a heatmap that depicts the modulation of gene expression in interferon-stimulated genes (ISGs) within peripheral blood mononuclear cells (PBMCs) harvested from wildtype and HuIFNAR mice. Cells were subjected to an in vitro stimulation period of 8 or 20 hours using either a saline solution (PBS; mock), human IFN α 2 (huIFN α 2), or mouse IFN α 5 (mIFN α 5). The color of the heatmap is determined by the normalized counts per million (CPM) values obtained from edgeR, providing a detailed representation of the differential gene expression across different treatment groups.

We understand your concerns about the inferences from native mouse IFN α in interpreting the data. We have analyzed the mouse IFN α gene expression profile changes (table 1, from the mRNA sequencing data related to Figure 3). We found that huIFN α 2 treatment doesn't induce mouse IFN α expression in most tissues except in the spleen. Even in the spleen, the mouse IFN α expression levels are extremely low. In addition, we have also analyzed the mouse IFN α expression to long-term huIFN α stimulation in the liver (table 2, from the mRNA sequencing data related to Figure 4). We don't observe an elevated expression of mouse IFN α in the liver. Therefore, the feedback regulation of huIFNAR by endogenous mouse IFN α expression can be maximally excluded in this regard.

Table 1. mouse FN a subtype expression in different tissues against huFN a2 stimulation

Tissue	Gene	huFNAR mouse mock treatment (reads/sam p le)			huFNAR mouse PegFN a2 treatment (reads/sam p le)			Tissue	Gene	huFNAR mouse mock treatment (reads/sam p le)			huFNAR mouse PegFN a2 treatment (reads/sam p le)				
		E51	E52	E53	E21	E22	E23			E51	E52	E53	E21	E22	E23		
b bod	lha1	0	0	0	0	0	0	heart	lha1	0	0	0	0	0	0		
	lha2	0	0	0	0	0	0		lha2	0	0	0	0	0	0		
	lha4	0	0	0	0	0	0		lha4	0	0	0	0	0	0		
	lha5	0	0	0	0	0	0		lha5	0	0	0	0	0	0		
	lha6	0	0	0	0	0	0		lha6	0	0	0	0	0	0		
	lha7	0	0	0	0	0	0		lha7	0	0	0	0	0	0		
	lha9	0	0	0	0	0	0		lha9	0	0	0	0	0	0		
	lha11	0	0	0	0	0	0		lha11	0	0	0	0	0	0		
	lha12	0	0	0	0	0	0		lha12	0	0	0	0	0	0		
	lha13	0	0	0	0	0	0		lha13	0	0	0	0	0	0		
	lha14	0	0	0	0	0	0		lha14	0	0	0	0	0	0		
	lha15	0	0	0	0	0	0		lha15	0	0	0	0	0	0		
	lha16	0	0	0	0	0	0		lha16	0	0	0	0	0	0		
	liver	lha1	0	0	0	0	0		0	m uscle	lha1	0	0	0	0	0	0
		lha2	0	0	0	0	0		0		lha2	0	0	0	0	0	0
		lha4	0	0	0	0	0		0		lha4	0	0	0	0	2	0
lha5		0	0	0	0	0	0	lha5	0		0	0	1	0	0		
lha6		0	0	0	0	0	0	lha6	0		0	0	0	0	0		
lha7		0	0	0	0	0	0	lha7	0		0	0	0	0	0		
lha9		0	0	0	0	0	0	lha9	0		0	0	0	0	0		
lha11		0	0	0	1	0	0	lha11	0		0	0	0	0	0		
lha12		0	0	0	0	0	0	lha12	0		0	0	0	0	0		
lha13		0	0	0	0	0	0	lha13	0		0	0	0	0	0		
lha14		0	0	0	0	0	0	lha14	0		0	0	0	0	0		
lha15		0	0	0	0	0	0	lha15	0		0	0	0	0	0		
lha16		0	0	0	0	0	0	lha16	0		0	0	0	0	0		
sp lren		lha1	0	0	0	8	2	5	i ntestine		lha1	0	0	0	0	0	0
		lha2	0	0	0	16	16	24			lha2	0	0	0	0	0	0
		lha4	0	0	0	5	7	10			lha4	0	0	0	0	0	0
	lha5	0	0	0	14	22	16	lha5		0	0	0	0	0	0		
	lha6	0	0	0	2	2	5	lha6		0	0	0	0	0	0		
	lha7	0	0	0	2	0	0	lha7		0	0	0	0	0	0		
	lha9	0	0	0	6	9	11	lha9		0	0	0	0	0	0		
	lha11	0	0	0	2	1	5	lha11		0	0	0	0	0	0		
	lha12	0	0	0	6	5	4	lha12		0	0	0	0	0	0		
	lha13	0	0	0	5	1	2	lha13		0	0	0	0	0	0		
	lha14	0	0	0	1	1	3	lha14		0	0	0	0	0	0		
	lha15	0	0	0	5	2	3	lha15		0	0	0	0	0	0		
	lha16	0	0	0	5	3	1	lha16		0	0	0	0	0	0		
	lung	lha1	0	0	0	0	1	0		bra in	lha1	0	0	0	0	0	0
		lha2	0	0	0	0	0	0			lha2	0	0	0	0	0	0
		lha4	0	0	0	0	0	0			lha4	0	0	0	0	2	0
lha5		0	1	0	0	0	0	lha5	0		0	0	0	0	0		
lha6		0	0	0	0	0	0	lha6	0		0	0	0	0	0		
lha7		0	0	0	0	0	0	lha7	0		0	0	0	0	0		
lha9		0	0	0	0	0	0	lha9	0		0	0	0	0	0		
lha11		0	0	0	0	0	0	lha11	0		0	0	0	0	0		
lha12		0	0	0	0	0	0	lha12	0		0	0	0	0	0		
lha13		0	0	0	0	0	0	lha13	0		0	0	0	0	0		
lha14		0	0	0	0	0	0	lha14	0		0	0	0	0	0		
lha15		0	0	0	0	0	0	lha15	0		0	0	0	0	0		
lha16		0	0	0	0	0	0	lha16	0		0	0	0	0	0		
kidney		lha1	0	0	0	0	0	0									
		lha2	0	0	0	0	0	0									
		lha4	0	0	0	0	0	1									
	lha5	0	0	0	0	0	0										
	lha6	0	0	0	0	0	0										
	lha7	0	0	0	0	0	0										
	lha9	0	0	0	0	0	0										
	lha11	0	0	0	0	0	0										
	lha12	0	0	0	0	0	0										
	lha13	0	0	0	0	0	0										
	lha14	0	0	0	0	0	0										
	lha15	0	0	0	0	0	0										
	lha16	0	0	0	0	0	0										

Table 2. mouse IFN α subtype expression in liver against long-term huIFN α 2 stimulation

Tissue	Gene	huIFNAR mouse mock treatment (reads/sample)					huIFNAR mouse PegIFN α 2 treatment (reads/sample)					
		CA2	CA5	CA6	CA10	CA13	CA1	CA3	CA4	CA7	CA9	CA11
liver (15 week huIFN α 2 treatment)	ifna1	0	0	0	0	0	0	0	0	0	0	0
	ifna2	0	0	0	0	0	0	0	0	0	0	0
	ifna4	0	0	0	0	0	0	0	0	0	0	0
	ifna5	0	0	0	0	0	0	0	0	0	0	0
	ifna6	0	0	0	0	0	0	0	0	0	0	0
	ifna7	0	0	0	0	0	0	0	0	0	0	0
	ifna9	0	0	0	0	0	0	0	0	0	0	0
	ifna11	0	0	0	0	0	0	0	0	0	0	0
	ifna12	0	0	0	0	0	0	0	0	0	0	0
	ifna13	0	0	0	0	0	0	0	0	0	0	0
	ifna14	0	0	0	0	0	0	0	0	0	0	0
	ifna15	0	0	0	0	0	0	0	0	0	0	0
	ifna16	0	0	0	0	0	0	0	0	0	0	0

Lastly, we want to express our heartfelt appreciation again. With your suggestion, our manuscript get well-improved.

2. The authors compared the effect of transgenic mouse PBMCs after in vivo Peg-IFN injection (10 μ g/mouse s.c.) and in vitro/ex vivo Peg-IFN stimulated (0.8 μ g/ml) human PBMCs. I would appreciate a direct comparison of transgenic mouse PBMCs and human PBMCs in vitro with the same Peg-IFN concentration. Maybe also a kinetic of early and late expressed ISGs (e.g., 6h vs. 18-24h) in in vitro IFN-stimulated human and mouse PBMCs might be of interest. How were the Peg-IFN concentrations selected for the different experimental conditions (in vitro/ex vivo vs. in vivo). In particular during AAV-HBV infection experiment, multiple injections of 2 μ g per mouse were performed, whereas in the initial in vivo experiment 10 μ g/mouse was injected.

Response: Thanks for your valuable comments about the IFN dose difference.

1) Frankly, at the beginning of this big project, we were worried about whether the huIFNAR mice could respond to human IFN α 2 treatment. Therefore, we used a way too high dose (10 μ g/ml) to test the responsiveness of huIFNAR for the first time when we got homozygous huIFNAR mice. We know the dosage is too high. Then, we learn that low doses also generate good stimulation, which gives us confidence.

In humanized mice, HBV-infected mice were treated for 12 weeks with 2 mg/kg of entecavir daily and/or with 30 μ g/kg of PEG-IFN twice weekly (Antimicrob Agents Chemother . 2017 Aug 24;61(9):e00725-17. doi: 10.1128/AAC.00725-17). In this reference, 30 μ g/kg of PEG-IFN twice weekly equals 2 μ g/mouse weekly. We choose

2ug/mouse in the following study. In this resubmission, to be consistent, we repeated the experiment with IFNa at a dose of 2ug/mouse (Fig. 1H-J).

We miscalculated the activity Unit of human PEG-IFNa2, used a higher concentration of human PEG-IFNa2 (0.8ug/ml) than usual for the first experiment and used that concentration afterwards. Fortunately, the 0.8ug/ml works in the in vitro experiment.

2) Your suggestion that “Maybe also a kinetic of early and late expressed ISGs (e.g., 6h vs. 18-24h) in in vitro IFN-stimulated human and mouse PBMCs might be of interest” is valuable. Unfortunately, the human ethics we used previously expired last year. Therefore, we have no resources to obtain PBMCs from healthy volunteers.

We understand your main idea is to observe the different kinetics of ISGs between human and mice after being stimulated with their counterpart. The difference can be addressed in an alternative manner. We have observed that the kinetic of early (8h) and late (20h) expressed ISGs in the wildtype C57BL/6 and huIFNAR mice differed (now provided as supplementary figure 6 in this revision). Please refer to the response to comment 1 for details.

Thanks again for your comment.

3. The authors analyzed very precisely the expression of the human IFNAR2 in various tissues. As the authors later also focused on various immune cell subsets, they should also analyze the expression of IFNAR1 and IFNAR2 in different immune cell subsets like T cells, NK cells, neutrophils, myeloid cells, and B cells.

Response: Thank you for your recognition of the IFNAR2 expression in various tissues. In Figure 3, the bulk sequencing of total mRNA included all the exons and can be used to calculate the full-length mRNA. We can analyze the expression of knock-in huIFAR2 by aligning the reads against the knock-in sequence.

In Figures 5 and 7, we used 3' sequencing, not full-length mRNA sequencing, to profile the intrahepatic single-cell characteristics after huIFNa long-term treatment sequencing. However, we should point out that the knock-in sequence of huIFNAR (as indicated in Figure 1A, and the huIFNAR sequence is supplied as Appendix 1 in the supplementary information file) is codon-optimized and is not in the alignment database for single-cell sequencing analysis. Therefore, as you commented, we cannot get the humanized IFNAR1 and R2 expression in different immune cell subsets.

But, for reviewing purposes, we find the endogenous mouse IFNAR1 and R2 gene expression from an online resource in respond to your comment. Unfortunately, we cannot change their formats. I hope this information will be helpful.

[REDACTED]

Figure 2 for review: Cell-specific mouse *Ifnar1* expression in different tissue in wildtype mouse. Modified from online database (<https://bis.zju.edu.cn/MCA/search2.html>).

[REDACTED]

Figure 3 for review: Cell-specific mouse Ifnar2 expression in different tissue in wildtype mouse. Modified from online database (<https://bis.zju.edu.cn/MCA/search2.html>).

4. Did the Peg-IFN-treated mice develop any side effects like fever or cytokine storm? As only one mouse (#3) achieved HBsAg and HBeAg loss during treatment, different concentrations of Peg-IFN should be considered in vivo.

Response: Thanks for your valuable comments.

1) Unfortunately, we did not monitor the mouse temperate throughout the experiment. Per your suggestion, we have stained the liver tissue with hematoxylin and eosin (HE) and asked an experienced pathologist to do the analysis. No liver pathology was observed in the IFNa2-treated huIFNAR mouse. One representative staining from each group is shown.

Figure 4 for review: Representative photomicrographs of liver sections stained with hematoxylin and eosin (HE). The liver in all three groups liver all showing normal hepatocytes arranged in cords, obvious sinusoids, and central vein.

2) Per your suggestion, we measure the serum proinflammatory cytokines using the 13-plex FACS-ELISA, including IL-23, IL-1 α , IFN- γ , TNF- α , CCL2 (MCP-1), IL-12p70, IL-1 β , IL-10, IL-6, IL-27, IL-17A, IFN- β and GM-CSF (BioLegend, cat# 740446). Fiver time points, including weeks 0, 1, 5, 10, and 14, are selected for the analysis because enough volume of plasma was left for analysis. Of the thirteen cytokines analyzed, none were significantly altered after PEG-IFN α 2 treatment.

We observe a seemly higher serum proinflammatory cytokines in the #3 mouse (red dot). Nevertheless, we have yet to conclude due to only one mouse.

The kinetics of proinflammatory cytokines is included in this revision as **supplementary Figure 11**. We hope the kinetics of proinflammatory cytokines benefit the understanding of the HBsAg loss.

Figure 5 for review: Inflammatory cytokine levels in plasma. Inflammation cytokines were tested by using Mouse Inflammation Panel (13-plex) with V-bottom Plate at 0, 1, 5, 10 and 14w.p.i. Bars represent the mean concentration of biological replicates \pm SEM, p values determined by one-way ANOVA. Red dot, the #3 mice in the main figure 4F.

We totally agree with your suggestion to test different concentrations of Peg-IFN since only one mouse in the huIFN α -treated achieved HBsAg seroconversion. We are working on this. Unfortunately, we need time to explore the methods, such as lower AAV-HBV concentration, to generate AAV-HBV mice with lower HBsAg levels. This study would be in another subsequent publication since at least one year is required to finish this new project. .

5. The data suggest that Peg-IFN treatment strongly affected CD8⁺ T cell responses during HBV infection. Apart from the extensive gene expression analysis, flow cytometry analysis of T effector cells showing cytotoxicity (GzmB, GzmA, perforin, CD107a, TRAIL, Fas, etc) or cytokine production (IL-2, TNF α , IFN γ , etc) should be added to the study. Additionally, the analysis of core/env-specific T cells by tetramer staining or in vitro peptide stimulation after treatment with PEG-IFN might be included to the manuscript. Furthermore, CD8 T cells during Peg-IFN treatment should be analyzed in depletion experiments to elucidate their exact role during IFN-therapy.

Response: Thanks for your valuable suggestions.

1) We completely agree with you that an additional flow cytometry analysis of T effector cells showing cytotoxicity or cytokine production will be helpful as a further corroboration of the gene expression analysis. The core/env-specific T cells by tetramer staining or in vitro peptide stimulation after treatment with PEG-IFN will provide more information. We attempted to obtain HBV-specific tetramers, but they are commercially unavailable in mainland China. We tried to measure HBV virus-specific T cell response to stimulation with 20-mer peptide pools (ELISPOT). Unfortunately, we only got negative results. As a control, we can get positive staining spots When using human PBMC from chronic HBV-infected individuals. Now this ELISPOT analysis is included in this reversion as supplementary figure 21.

Figure 6 for review: ELISpot analysis of IFN γ secretion in mouse splenocytes as indicated. Splenocytes were collected after 15 weeks after IFN α 2 injected and the number of IFN-c secreting cells generated in response to S1,S2 and core peptides. (A) Sample wells with spots with mock or HBV peptide stimulation. Each condition was performed in duplicates. (B) Frequencies of HBV peptide-specific spots per million

splenocytes in WT, MOCK, or IFN2 α mice. Data in are means \pm SEM , p values were determined by one-way ANOVA.

3) Your suggestion of CD8+ depletion is very valuable. Actually, we have brought an acting model about the mechanism of the huIFNa treatment on HBV. We would like to share our hypothesis here for the review purpose only.

Figure 7 for review: Hypothesis of the axis of CCL2 and CD8 T cells in reducing HBsAg.

Hypothesis: Based on our current study and observed clinical data, we propose that interferon exerts its effect on HBV infection through the “CCL2/CCR2-Ly6c+ monocytes-CD8+ effector memory T cell” axis. Specifically, IFN- α enhances the expression of CCL2/CCR2. This upregulation triggers pro-inflammatory changes in macrophages towards an M1-like phenotype. Simultaneously, it also promotes the intrahepatic recruitment of monocytes/macrophages, which act as a crucial bridge connecting innate and adaptive immunity. Subsequently, Ly6c+ monocytes present antigens to CD8+ T cells within the liver, fostering an increase in CD8+ effector memory T cells (CD8+TEM) in the hepatic environment. This process ultimately contributes to the anti-HBV effect.

We are in the middle of a grant application. We are eager to explore the underlying mechanism. Before formally initiating the program, we are still working on the detailed experimental design. There are several key technical challenges.

As we all know, HBV-related experiments in vivo always take a much long-time (in the current study, it takes over 20 weeks, nearly half a year). In order to deplete mCCL2, what antibody concentration should be used if we use the mCCL2-depleting antibody? Is it better and more convincing if we generate a complete mCCL2 knockout mouse on top

of the huIFNAR mouse? Because mouse CCL2 are continuously produced by various types of cells and cytokines depletion using antibodies is usually insufficient, we are prone to make the CCL2 knockout mouse.

For the CD8 + cell depletion, the experiment is straightforward. Cell population depletion by depleting antibodies is an efficient way. We are prone to an antibody depletion strategy to deplete CD8 T in further study. Depleting antibody titration is required before we start the experiment. We will focus on this part and keep these in our next publication.

Due to the limited space. Sorry for not being able to include this part in the current submission. Please allow us to keep the CD8+ depletion analysis to our ongoing project. We will also submit the study to Nature Communications when it is done. Thanks.

6. Please change the order of the figures: Fig.7 is mentioned in the text before Fig. 6 and Supp.Fig.14 and Supp. Fig.15 before Supp. Fig.13.

Response: Thanks. We have made the figures in the right order. Also, we have re-organized the main figures from seven to eight, expanded the supplementary figures from fifteen to twenty-one, and included two appendix information for mouse generation and knock-in sequencing confirmation.

Reviewer #2 (Remarks to the Author):

Reviewer comments on “Comprehensive immune profile changes to human interferon stimulation in normal and HBV-infected conditions in a humanized mouse”.

The authors have established a human IFNAR knock-in mouse model which they tested using in vivo and ex vivo treatment with human IFN α . Furthermore, they applied the IFN treatment in the context of the AAV-HBV model. They also performed bulk and single-cell RNA-Seq analysis and also looked into specific ISGs using qRT-PCR. Despite the extensive work that is presented here, essential controls are missing that would highlight the importance for this mouse model and its application in IFN-related studies including treatment of HBV. Below there are specific comments figure by figure for consideration by the authors.

Response: Thank you for your constructive suggestions. We have diligently conducted all the experiments possible in accordance with your specific comments.

1. In Figure 1 E-H it would be more helpful to include a WB for mMx2 or mISG15. This should be quite easy since these are ex vivo samples from WT and HuIFNAR mice stimulated with human IFN α 2. An essential control for this experiment is also treatment of the cells with murine IFN α . This will also allow for a comparison between the two systems but also confirm that there is no mouse-dependent stimulation in HuIFNAR. In addition, in Figure 1 I-J injection of murine IFN α would serve as a good control with both mRNA and protein levels of mMX1 and mISG15 shown. Is there a reason for switching from Mx2 in panels E and G to Mx1 in panel I? In panel 1C I would also suggest showing protein levels of HuIFNAR.

Response: Thank you for your valuable suggestions. We make changes accordingly in this revision. We know your main concern is whether the current model can generate a specific response to human IFN α . We make the following changes to address your and the other reviewers' concerns.

1) We detected the splenic mISG15 protein level changes against mIFN α 5 and huIFN α 2 stimulation in vivo. Compared to the mock-treated wildtype mice, huIFN α 2 treatment did not induce mISG15 expression in wildtype mice, while mIFN α induced an elevated mISG15 expression. However, in the huIFNAR mice, huIFN α 2 stimulation led to an obvious increase in mISG15 expression than mIFN α 5 ($p=0.0013$). The WB of mISG15 is now included as the supplementary Figure 5 in this revision.

Figure 1 for review: Western blot analysis of Isg15 proteins. (A) Mice were subcutaneously injected with huIFNα2 and/or mIFNα5. Isg15 levels were analyzed using Western blotting 16 hours post-injection (Blots were cropped and presented). (B) The bar graph represents a densitometric plot of mIsg15 protein expression across different groups. Data in are means ± SEM , p values determined by one-way ANOVA.

2) We test whether the human IFNAR antagonist antibody (anti-R2 and anti-R1), newly developed antibodies that can bind to the huIFNAR receptor, could block the human IFN activity in huIFNR cells. PBMCs were isolated and treated as indicated in vitro (Fig. 1G). We observed a specific inhibition of human IFNα2 stimulation.

Therefore, the above experiments demonstrated that the huIFNAR mouse can respond specifically to human IFNα.

3) Following your suggestion, we assessed the responsiveness of the huIFNAR mouse to mouse stimulation and selected mouse IFNα5 due to its ready availability. Mouse PBMCs isolated from wild-type C57BL/6C and huIFNAR mice were stimulated with huIFNα2 and mIFNα5 for 8h and 20h in vitro as indicated. The ISG gene expression changes among groups were shown in the heatmap (now provided as supplementary figure 6). We observed that human IFNα2 could not evoke obvious gene expression in mouse PBMCs compared to the mouse IFNα5.

As for the huIFNAR mouse PBMCs, human IFNα2 simulated a different pattern from what was observed in the wildtype mouse PBMC. Meanwhile, huIFNAR mouse PBMCs also responded to mIFNα5 stimulation. Although likely, the exaction pattern and magnitude differ substantially at 8h and 20h post-stimulation. As the wildtype mouse IFNAR1 submit is still intact in our design, we postulated that the mouse IFNAR1 and humanized IFNAR2 can form dimmers and maintain their response to mouse IFNα stimulation.

Your valuable comments make us determined to further knockout the mouse IFNAR1 on top of the current huIFNAR mouse. We have started to generate this mouse recently. But it will take at least one year to get a colony of homozygous huIFNAR and mouse IFNAR1 knockout mice to test this postulation.

Figure 2 for review: Transcriptomic Response to Human and Mouse interferons in Wildtype and HuIFNAR Mice. The figure illustrates a heatmap that depicts the modulation of gene expression in interferon-stimulated genes (ISGs) within peripheral blood mononuclear cells (PBMCs) harvested from wildtype and HuIFNAR mice. Cells were subjected to an in vitro stimulation period of 8 or 20 hours using either a saline solution (PBS; mock), human IFN α 2 (huIFN α 2), or mouse IFN α 5 (mIFN α 5). The color of the heatmap is determined by the normalized counts per million (CPM) values obtained from edgeR, providing a detailed representation of the differential gene expression across different treatment groups.

We understand your concerns about the inferences from native mouse IFN α in interpreting the data. We have analyzed the mouse IFN α gene expression profile changes (data related to Figure 3). We found that huIFN α 2 treatment doesn't induce mouse IFN α expression in most tissues except in the spleen. Even in the spleen, the mouse IFN α expression levels are extremely low. In addition, we have also analyzed the mouse IFN α expression to long-term huIFN α stimulation in the liver (data related to

Figure 4). We don't observe an elevated expression of mouse IFN α in the liver. Therefore, the feedback regulation of huIFNAR by endogenous mouse IFN α expression can be maximally excluded in this regard.

Table 1. mouse IFN α subtype expression in different tissues against huIFN α 2 stimulation

Tissue	Gene	huFNAR mouse mock treatment (reads/sam ple)			huFNAR mouse PegFN α 2 treatment (reads/sam ple)			Tissue	Gene	huFNAR mouse mock treatment (reads/sam ple)			huFNAR mouse PegFN α 2 treatment (reads/sam ple)				
		E51	E52	E53	E21	E22	E23			E51	E52	E53	E21	E22	E23		
blood	Ifna1	0	0	0	0	0	0	heart	Ifna1	0	0	0	0	0	0		
	Ifna2	0	0	0	0	0	0		Ifna2	0	0	0	0	0	0		
	Ifna4	0	0	0	0	0	0		Ifna4	0	0	0	0	0	0		
	Ifna5	0	0	0	0	0	0		Ifna5	0	0	0	0	0	0		
	Ifna6	0	0	0	0	0	0		Ifna6	0	0	0	0	0	0		
	Ifna7	0	0	0	0	0	0		Ifna7	0	0	0	0	0	0		
	Ifna9	0	0	0	0	0	0		Ifna9	0	0	0	0	0	0		
	Ifna11	0	0	0	0	0	0		Ifna11	0	0	0	0	0	0		
	Ifna12	0	0	0	0	0	0		Ifna12	0	0	0	0	0	0		
	Ifna13	0	0	0	0	0	0		Ifna13	0	0	0	0	0	0		
	Ifna14	0	0	0	0	0	0		Ifna14	0	0	0	0	0	0		
	Ifna15	0	0	0	0	0	0		Ifna15	0	0	0	0	0	0		
	Ifna16	0	0	0	0	0	0		Ifna16	0	0	0	0	0	0		
	liver	Ifna1	0	0	0	0	0		0	muscle	Ifna1	0	0	0	0	0	0
		Ifna2	0	0	0	0	0		0		Ifna2	0	0	0	0	0	0
		Ifna4	0	0	0	0	0		0		Ifna4	0	0	0	0	2	0
Ifna5		0	0	0	0	0	0	Ifna5	0		0	0	1	0	0		
Ifna6		0	0	0	0	0	0	Ifna6	0		0	0	0	0	0		
Ifna7		0	0	0	0	0	0	Ifna7	0		0	0	0	0	0		
Ifna9		0	0	0	0	0	0	Ifna9	0		0	0	0	0	0		
Ifna11		0	0	0	1	0	0	Ifna11	0		0	0	0	0	0		
Ifna12		0	0	0	0	0	0	Ifna12	0		0	0	0	0	0		
Ifna13		0	0	0	0	0	0	Ifna13	0		0	0	0	0	0		
Ifna14		0	0	0	0	0	0	Ifna14	0		0	0	0	0	0		
Ifna15		0	0	0	0	0	0	Ifna15	0		0	0	0	0	0		
Ifna16		0	0	0	0	0	0	Ifna16	0		0	0	0	0	0		
spleen		Ifna1	0	0	0	8	2	5	intestine		Ifna1	0	0	0	0	0	0
		Ifna2	0	0	0	16	16	24			Ifna2	0	0	0	0	0	0
		Ifna4	0	0	0	5	7	10			Ifna4	0	0	0	0	0	0
	Ifna5	0	0	0	14	22	16	Ifna5		0	0	0	0	0	0		
	Ifna6	0	0	0	2	2	5	Ifna6		0	0	0	0	0	0		
	Ifna7	0	0	0	2	0	0	Ifna7		0	0	0	0	0	0		
	Ifna9	0	0	0	6	9	11	Ifna9		0	0	0	0	0	0		
	Ifna11	0	0	0	2	1	5	Ifna11		0	0	0	0	0	0		
	Ifna12	0	0	0	6	5	4	Ifna12		0	0	0	0	0	0		
	Ifna13	0	0	0	5	1	2	Ifna13		0	0	0	0	0	0		
	Ifna14	0	0	0	1	1	3	Ifna14		0	0	0	0	0	0		
	Ifna15	0	0	0	5	2	3	Ifna15		0	0	0	0	0	0		
	Ifna16	0	0	0	5	3	1	Ifna16		0	0	0	0	0	0		
	lung	Ifna1	0	0	0	0	1	0		brain	Ifna1	0	0	0	0	0	0
		Ifna2	0	0	0	0	0	0			Ifna2	0	0	0	0	0	0
		Ifna4	0	0	0	0	0	0			Ifna4	0	0	0	0	2	0
Ifna5		0	1	0	0	0	0	Ifna5	0		0	0	0	0	0		
Ifna6		0	0	0	0	0	0	Ifna6	0		0	0	0	0	0		
Ifna7		0	0	0	0	0	0	Ifna7	0		0	0	0	0	0		
Ifna9		0	0	0	0	0	0	Ifna9	0		0	0	0	0	0		
Ifna11		0	0	0	0	0	0	Ifna11	0		0	0	0	0	0		
Ifna12		0	0	0	0	0	0	Ifna12	0		0	0	0	0	0		
Ifna13		0	0	0	0	0	0	Ifna13	0		0	0	0	0	0		
Ifna14		0	0	0	0	0	0	Ifna14	0		0	0	0	0	0		
Ifna15		0	0	0	0	0	0	Ifna15	0		0	0	0	0	0		
Ifna16		0	0	0	0	0	0	Ifna16	0		0	0	0	0	0		
kidney		Ifna1	0	0	0	0	0	0									
		Ifna2	0	0	0	0	0	0									
		Ifna4	0	0	0	0	0	1									
	Ifna5	0	0	0	0	0	0										
	Ifna6	0	0	0	0	0	0										
	Ifna7	0	0	0	0	0	0										
	Ifna9	0	0	0	0	0	0										
	Ifna11	0	0	0	0	0	0										
	Ifna12	0	0	0	0	0	0										
	Ifna13	0	0	0	0	0	0										
	Ifna14	0	0	0	0	0	0										
	Ifna15	0	0	0	0	0	0										
	Ifna16	0	0	0	0	0	0										

Table 2. mouse IFN α subtype expression in liver against long-term huIFN α 2 stimulation

Tissue	Gene	huIFNAR mouse mock treatment (reads/sample)					huIFNAR mouse PegIFN α 2 treatment (reads/sample)					
		CA2	CA5	CA6	CA10	CA13	CA1	CA3	CA4	CA7	CA9	CA11
liver (15 week huIFN α 2 treatment)	ifna1	0	0	0	0	0	0	0	0	0	0	0
	ifna2	0	0	0	0	0	0	0	0	0	0	0
	ifna4	0	0	0	0	0	0	0	0	0	0	0
	ifna5	0	0	0	0	0	0	0	0	0	0	0
	ifna6	0	0	0	0	0	0	0	0	0	0	0
	ifna7	0	0	0	0	0	0	0	0	0	0	0
	ifna9	0	0	0	0	0	0	0	0	0	0	0
	ifna11	0	0	0	0	0	0	0	0	0	0	0
	ifna12	0	0	0	0	0	0	0	0	0	0	0
	ifna13	0	0	0	0	0	0	0	0	0	0	0
	ifna14	0	0	0	0	0	0	0	0	0	0	0
	ifna15	0	0	0	0	0	0	0	0	0	0	0
	ifna16	0	0	0	0	0	0	0	0	0	0	0

To be consistent, we include mMx1 and mISG15 as the representative ISGs to reflect their response to IFN stimulation under different circumstances.

We also include the western blot detection of mISG15 in vivo as a supplementary figure to make the manuscript more informative.

From the design, you can see that we make a hybrid chimeric huIFNAR, consisting of both extracellular moieties from huIFNAR and intracellular moieties from mouse IFNAR.

2. In Figure 2A/Supplementary Figure S4 the experimental design would have been better controlled if the murine PBMCs were treated with IFN α after isolation similar to what happened with the human PBMCs. Maybe it would be best to include a third group that is similar to how the stimulation was done in human PBMCs. In Figure 2B these numbers exclude any ISGs that are not found in both humans and mice? This was not clear in the text.

Response: Thanks for your suggestions.

1) We have made a detailed explanation above in response to comment 1. Our newly added mRNA sequencing analysis will benefit the understanding of this novel mouse's necessity and add clarity to this manuscript.

2) The numbers of ISGs vary markedly among different species and exhibit tissue-specific characteristics. The following paragraph is copied from a review paper published in Annual Review of Virology With the title "Interferon-Stimulated Genes: What Do They All Do?" (Annu Rev Virol . 2019 Sep 29;6(1):567-584).

Early microarray studies suggested the existence of >300 ISGs. Our meta-analysis of ISG transcriptomes published before 2007 indicated that approximately 450 genes were commonly induced by type I IFNs across multiple cell backgrounds from diverse mammals (29). Notably, cells of hematopoietic lineage induce many more genes, nearly 1,000 in chimpanzee or mouse cells treated with IFN α (17, 28). We now know that the total number of ISGs may be even higher. Indeed, in a recent study, Shaw et al. (23) used RNA-Seq and cross-comparative analysis to profile the IFN-induced transcriptional response in primary fibroblasts from nine diverse mammals and one bird. In human cells, approximately 10% of the human genome was subject to regulation by IFN. Notably, whereas most other studies implemented a fold change cutoff to designate ISGs, the authors of this study implemented only a statistical cutoff based on false discovery rate. For example, if a particular gene was reproducibly expressed 1.3-fold higher in the presence of IFN, that gene was considered an ISG. In most other studies, such genes would likely not be considered ISGs because they would fall below a preset 1.5-fold, 2-fold, or even 4-fold induction cutoff. Thus, while the total number of genes induced by IFN is very high, many of these genes have very low levels of induction, which may make it difficult to determine whether they are bona fide antiviral ISGs. Notably, when Shaw et al. (23) compared all ISG transcriptomes from diverse animals, they found only 62 common genes, suggesting that this core ISG set represents an evolutionarily conserved group of ancestral genes critical to host IFN responses.

In our study, we choose a list of human ISGs (487 genes) for analysis. Only those ISGs with significant changes were shown in Figure 2B.

3. In Figure 3 to better appreciate the value of these data I would propose to include as a control treatment with mouse IFN α in a WT mouse. If the results are identical then what new did we learn from this experiment/model?

Response: Thanks for your valuable suggestions.

We understand your concern about whether human IFN α stimulated a different immune response compared with the murine IFN α in wildtype mice. We have shown the huIFNAR mice generate varied gene expression profiles against mIFN α 5 and huIFNAR (newly added in supplementary Figure 6). This new data benefit the audience.

4. Line 267: FigS7 is probably FigS8. The AAV HBV 1.3X is used in HBV research but since there is no cccDNA present in the mouse hepatocytes, modeling chronic HBV

infection is incomplete. This is an inherent drawback of this approach and not the manuscript's but it should be mentioned for clarification.

Response: The mislabels are corrected.

Thank you for your understanding this AAV-HBV model's inherent drawback.

In the circular HBV cccDNA template, only four promoters regulate the 5 HBV mRNA transcription from the overlapped HBV cccDNA. The HBV cccDNA can transform into supercoiled mini-chromosome. These 4 promoters might affect each other in a well-coordinated and synergistic manner. For example, the inhibition of promoters 2 and 3 might also directly inhibit the downstream promoter 4 and promoter 1, which control the HBx and PreCore and Core mRNA transcription. The drop of HBsAg will likely result in the decline of HBx and HBx and HBV DNA levels.

HBV gene expression from HBV cccDNA

Figure 3 for review: HBV viral mRNA transcription and protein expression from HBV cccDNA template. Promoter 1, 2, 3 and 4 are indicated.

However, the AAV-HBV 1.3-fold genome vector contains six promoters, consisting of two promoter 1, one promoter 2, one promoter 3 and two promoter 4. In the same circumstance when promoters 2 and 3 are inhibited, the promoters 4 and 1 downstream (within the 1.0X HBV genome) might be suppressed too. But, the promoter 4 and the promoter 1 upstream (within the 0.3X genome) might still be active, and still be able to transcribe the HBx mRNA (0.7kb) and preCore and pre-genomic RNA (3.5 kb). Therefore, the HBsAg drop was not accompanied by an HBeAg and HBV DNA decline in the AAV-HBV 1.3X model.

There are still other models to explain the difference between AAV-HBV 1.3X and HBV cccDNA.

HBV gene expression from AAV-HBV 1.3X

Figure 4 for review: **HBV viral mRNA transcription and protein expression from AAV-HBV 1.3X template.** Promoter 1, 2, 3 and 4 are indicated.

5. Figure 4 should include a control with murine IFNa on C57BL/6C.

Figure 4F should include some information on the state of the hepatocytes to exclude toxicity due to IFN treatment. The starting point for HBsAg is also 1/3 of what is shown in panel 4B with the scale being modified.

Response: Thanks for your suggestion.

1) We completely agree with you that including a murine IFNa control on C57BL/6C will be helpful.

We once debated on the appropriateness of including murine IFNa on C57BL/6C as a control. The human PEG-IFNa2 is readily available from pharmaceutical resources. Pegylation modification of human IFNa2 stabilizes its function and substantially extends its half-life in vivo. Weekly application of human PEG-IFNa2 for many weeks is possible.

Nevertheless, a long-effective mouse IFNa is unavailable. Daily mouse IFNa injection for 15 weeks (>100 times) will be technically challenging for our experiment if applied. Also, the frequent daily injection doesn't match the weekly human IFNa2 injection.

In addition, high-quality mouse IFNa is on the shelf from PBL assay science. But frankly, their price is costly and unaffordable for long-term mouse experiments.

Therefore, considering the feasibility, we did not include this group in our analysis.

2) During the revision, we analyze the liver toxicity by staining the liver tissue with hematoxylin and eosin (HE). We also asked an experienced pathologist to do the analysis. No liver pathology was observed in the IFNa2-treated hulfNAR mouse. One representative staining from each group is shown. Now the HE staining is provided as a supplementary Figure 13.

Figure 5 for review : Representative photomicrographs of liver sections stained with hematoxylin and eosin (HE). The liver in all three groups liver all showing normal hepatocytes arranged in cords, obvious sinusoids, and central vein.

By the way, we measure the serum proinflammatory cytokines using the 13-plex FACS-ELISA, including IL-23, IL-1 α , IFN- γ , TNF- α , CCL2 (MCP-1), IL-12p70, IL-1 β , IL-10, IL-6, IL-27, IL-17A, IFN- β and GM-CSF (BioLegend, cat# 740446). Fiver time points, including weeks 0, 1, 5, 10, and 14, are selected for the analysis because enough volume of plasma was left for analysis. Of the thirteen cytokines analyzed, none were significantly altered after PEG-IFN α 2 treatment.

We observe a seemly higher serum proinflammatory cytokines in the #3 mouse (red dot). Nevertheless, we have yet to conclude due to only one mouse.

The kinetics of proinflammatory cytokines is included in this revision as supplementary Figure 14. We hope the kinetics of proinflammatory cytokines benefit the understanding of the HBsAg loss.

Figure 6 for review : ELISpot analysis of IFN γ secretion in mouse splenocytes

as indicated. Splenocytes were collected after 15 weeks after IFN α 2 injected and the number of IFN-c secreting cells generated in response to S1,S2 and core peptides. (A) Sample wells with spots with mock or HBV peptide stimulation. Each condition was performed in duplicates. (B) Frequencies of HBV peptide-specific spots per million splenocytes in WT, MOCK, or IFN2 α mice. Data in are means \pm SEM , p values were determined by one-way ANOVA.

3) Yes, the starting point for HBsAg in panel 4F is also 1/3 of what is shown in panel 4B. The scale change is to present the #3 mouse in an easier-read manner. Low HBsAg levels are associated with a high opportunity to HBV functional cure in clinical practice. In our study, the #3 mouse with the lowest HBsAg level among all the mice in the PEG-IFN α 2 treated cohort mimicked what was observed in clinical, despite that we cannot conclude due to the small mouse number.

The #3 mice in panel 4F encourage us to investigate the HBsAg concentrations dependent HBV functional cure by PEG-IFN α 2. We hope this investigation could provide insight into HBV clinical treatment.

6. Figure 5. The single cell data would benefit from an earlier time point because at week 15 there might be increased toxicity accumulated from IFN treatment.

Response: Thanks for your suggestion to include another earlier time point. We want to point out that single-cell sequencing analysis incurs substantial costs and is unfordable for us as a small laboratory. Conducting extensive analysis on a larger number of samples at multiple time points would require additional grant support. We kindly request your support and understanding in facilitating our grant application, which would enable us to conduct more comprehensive analyses in the future. We hope that you understand the challenges we are facing in this regard.

For some reason Fig. 7 is mentioned before Fig. 6 which is confusing.

Response: Thanks for your reminding. We have changed this mistake in the revision. The initial Fig.7 is divided into Fig.6 and Fig.8 in this revision.

Minor but would need to be addressed: English proofreading would make it easier to read some parts even though the majority of the manuscript is well-written.

Response: Thank you for your understanding. As non-native English speakers, we know our expression will inevitably need further changes before publication. As what we always have done in other publications, we have also ordered a language editing service from Springer Nature Author Service (order number: YRTLK9PF) to improve the expression. Their changes will benefit the clarity and effectiveness of our manuscript.

Reviewer #3 (Remarks to the Author):

The manuscript by Wang et al. aims to analyze the mechanisms of action of human IFN α to HBV via the human IFNAR by developing a model with a knock-in of the human *Ifnar* in the mouse *Ifnar2* locus.

The data sets need to be clarified, some experimental controls are missing and the conclusions are not substantiated by the experiments with too small cohorts. Some of the statements do not reflect the literature in the field and are forgetting important concepts.

Response: Thank you for your constructive suggestions. We have diligently conducted all the experiments possible in accordance with your specific comments.

The proper use of English: spelling, grammar, and incorrect word usage must be revised throughout the manuscript.

Response: We have ordered a Spring Nature Author service (order number: YRTLK9PF) to help us change language and hope the language is improved. Thanks.

1. The idea that IFN α therapy is essential to HBV functional cure is the authors' opinion and should not be stated as a fact, such sentences must be revised: page 6 line 112-114 "The IFN α -plus corroboration strategy on top of most HBV direct antivirals, subsequently or simultaneously, seems inevitable to achieve sustained HBV functional cure." HBV functional cure occurs more frequently without IFN α treatment than with it.

Response: Thanks for your comment.

HBV functional cure is defined as loss of hepatitis B surface antigen (HBsAg) and undetectable hepatitis B virus (HBV) DNA after six months off therapy; it is associated with improved clinical outcomes and is the optimal goal of therapy for chronic hepatitis B.

I want to explain the current situation of PEG-IFN treatment in HBV patients. I want to recommend the EASL guideline for HBV treatment (Title: EASL 2017 Clinical Practice Guidelines on the management of hepatitis B virus infection, [https://www.journal-of-hepatology.eu/article/S0168-8278\(17\)30185-X/fulltext](https://www.journal-of-hepatology.eu/article/S0168-8278(17)30185-X/fulltext)). From this guideline, we can see that HBsAg loss is pretty low in HBeAg positive and negative patients (3-7%). HBsAg loss is rare under nucleos(t)ide analogue (NA) mono-therapy.

Recently, IFN α has been recommended to treat NAs-treated CHB patients who have declined HBsAg with PEG-IFN α .

[REDACTED]

Figure 1 for review : Title: EASL 2017 Clinical Practice Guidelines on the management of hepatitis B virus infection, [https://www.journal-of-hepatology.eu/article/S0168-8278\(17\)30185-X/fulltext](https://www.journal-of-hepatology.eu/article/S0168-8278(17)30185-X/fulltext)

2. The species cross-reactivity of human to mouse IFN α needs to be clarified by in vitro and in vivo experiments, analysis of sequences as done Fig S1 and S2 is insufficient. Experiments showing the mouse IFNAR response (or lack thereof) to hIFN α 2 are required to justify the importance of establishing a humanized IFNAR mouse model.

Response: Thank you for your constructive suggestion. We understand you care more about the necessity of generating a huIFNAR mouse. If wildtype mouse could respond to human IFN α stimulation identically to humans, our study will be unnecessary.

Following your and the other two reviewer's suggestion, we assessed the responsiveness of the huIFNAR mouse to mouse stimulation and selected mouse IFN α 5 due to its ready availability. Mouse PBMCs isolated from wildtype C57BL/6C and huIFNAR mice were stimulated with huIFN α 2 and mIFN α 5 for 8 h and 20 h in vitro as indicated. The ISG gene expression changes among groups were shown in the heatmap (now provided as supplementary figure 6). We observed that human IFN α 2 could not evoke obvious gene expression in mouse PBMCs compared to the mouse IFN α 5.

Figure 2 for review: Transcriptomic Response to Human and Mouse interferons in Wildtype and HuIFNAR Mice. The figure illustrates a heatmap that depicts the modulation of gene expression in interferon-stimulated genes (ISGs) within peripheral blood mononuclear cells (PBMCs) harvested from wildtype and HuIFNAR mice. Cells were subjected to an in vitro stimulation period of 8 or 20 hours using either a saline solution (PBS; mock), human IFN α 2 (huIFN α 2), or mouse IFN α 5 (mIFN α 5). The color of the heatmap is determined by the normalized counts per million (CPM) values obtained from edgeR, providing a detailed representation of the differential gene expression across different treatment groups.

As for the huIFNAR mouse PBMCs, human IFN α 2 simulated a different pattern from the wildtype mouse PBMC. Meanwhile, huIFNAR mouse PBMCs also responded to mIFN α 5 stimulation. Although likely, the exaction pattern and magnitude differ substantially at both 8 h and 20 h post-stimulation. As the wildtype mouse IFNAR1 subunit remains intact in our design, we postulated that the mouse IFNAR1 and humanized IFNAR2 can form dimmers and maintain their response to mouse IFN α stimulation.

Your valuable comments make us determined to knock out further the mouse IFNAR1 on top of the current huIFNAR mouse. We have started to generate this mouse

recently. But it will take at least one year to get a colony of homozygous huIFNAR and mouse IFNAR1 knockout mice to test this postulation.

The new data raises another concern about the possible inferences from native mouse IFNa in interpreting the data. To address this question, we have analyzed the mouse IFNA gene expression profile changes (table 1, from the mRNA sequencing data related to Figure 3). We found that huIFNa2 treatment doesn't induce mouse IFNa expression in most tissues except in the spleen. Even in the spleen, the mouse IFNa expression levels are extremely low. In addition, we have also analyzed the mouse IFNa expression to long-term huIFNa stimulation in the liver (table 2, from the mRNA sequencing data related to Figure 4). We don't observe an elevated expression of mouse IFNa in the liver. Therefore, the feedback regulation of huIFNAR by endogenous mouse IFNa expression can be maximally excluded in this regard.

Table 1. mouse FN a subtype expression in different tissues against huFN a2 stimulation

Tissue	Gene	huFNAR mouse mock treatment (reads/sam p k)			huFNAR mouse PegFN a2 treatment (reads/sam p k)			Tissue	Gene	huFNAR mouse mock treatment (reads/sam p k)			huFNAR mouse PegFN a2 treatment (reads/sam p k)				
		E51	E52	E53	E21	E22	E23			E51	E52	E53	E21	E22	E23		
b bod	lha1	0	0	0	0	0	0	heart	lha1	0	0	0	0	0	0		
	lha2	0	0	0	0	0	0		lha2	0	0	0	0	0	0		
	lha4	0	0	0	0	0	0		lha4	0	0	0	0	0	0		
	lha5	0	0	0	0	0	0		lha5	0	0	0	0	0	0		
	lha6	0	0	0	0	0	0		lha6	0	0	0	0	0	0		
	lha7	0	0	0	0	0	0		lha7	0	0	0	0	0	0		
	lha9	0	0	0	0	0	0		lha9	0	0	0	0	0	0		
	lha11	0	0	0	0	0	0		lha11	0	0	0	0	0	0		
	lha12	0	0	0	0	0	0		lha12	0	0	0	0	0	0		
	lha13	0	0	0	0	0	0		lha13	0	0	0	0	0	0		
	lha14	0	0	0	0	0	0		lha14	0	0	0	0	0	0		
	lha15	0	0	0	0	0	0		lha15	0	0	0	0	0	0		
	lha16	0	0	0	0	0	0		lha16	0	0	0	0	0	0		
	liver	lha1	0	0	0	0	0		0	muscle	lha1	0	0	0	0	0	0
		lha2	0	0	0	0	0		0		lha2	0	0	0	0	0	0
		lha4	0	0	0	0	0		0		lha4	0	0	0	0	2	0
lha5		0	0	0	0	0	0	lha5	0		0	0	1	0	0		
lha6		0	0	0	0	0	0	lha6	0		0	0	0	0	0		
lha7		0	0	0	0	0	0	lha7	0		0	0	0	0	0		
lha9		0	0	0	0	0	0	lha9	0		0	0	0	0	0		
lha11		0	0	0	1	0	0	lha11	0		0	0	0	0	0		
lha12		0	0	0	0	0	0	lha12	0		0	0	0	0	0		
lha13		0	0	0	0	0	0	lha13	0		0	0	0	0	0		
lha14		0	0	0	0	0	0	lha14	0		0	0	0	0	0		
lha15		0	0	0	0	0	0	lha15	0		0	0	0	0	0		
lha16		0	0	0	0	0	0	lha16	0		0	0	0	0	0		
spleen		lha1	0	0	0	8	2	5	intestine		lha1	0	0	0	0	0	0
		lha2	0	0	0	16	16	24			lha2	0	0	0	0	0	0
		lha4	0	0	0	5	7	10			lha4	0	0	0	0	0	0
	lha5	0	0	0	14	22	16	lha5		0	0	0	0	0	0		
	lha6	0	0	0	2	2	5	lha6		0	0	0	0	0	0		
	lha7	0	0	0	2	0	0	lha7		0	0	0	0	0	0		
	lha9	0	0	0	6	9	11	lha9		0	0	0	0	0	0		
	lha11	0	0	0	2	1	5	lha11		0	0	0	0	0	0		
	lha12	0	0	0	6	5	4	lha12		0	0	0	0	0	0		
	lha13	0	0	0	5	1	2	lha13		0	0	0	0	0	0		
	lha14	0	0	0	1	1	3	lha14		0	0	0	0	0	0		
	lha15	0	0	0	5	2	3	lha15		0	0	0	0	0	0		
	lha16	0	0	0	5	3	1	lha16		0	0	0	0	0	0		
	lung	lha1	0	0	0	0	1	0		brain	lha1	0	0	0	0	0	0
		lha2	0	0	0	0	0	0			lha2	0	0	0	0	0	0
		lha4	0	0	0	0	0	0			lha4	0	0	0	0	2	0
lha5		0	1	0	0	0	0	lha5	0		0	0	0	0	0		
lha6		0	0	0	0	0	0	lha6	0		0	0	0	0	0		
lha7		0	0	0	0	0	0	lha7	0		0	0	0	0	0		
lha9		0	0	0	0	0	0	lha9	0		0	0	0	0	0		
lha11		0	0	0	0	0	0	lha11	0		0	0	0	0	0		
lha12		0	0	0	0	0	0	lha12	0		0	0	0	0	0		
lha13		0	0	0	0	0	0	lha13	0		0	0	0	0	0		
lha14		0	0	0	0	0	0	lha14	0		0	0	0	0	0		
lha15		0	0	0	0	0	0	lha15	0		0	0	0	0	0		
lha16		0	0	0	0	0	0	lha16	0		0	0	0	0	0		
kidney		lha1	0	0	0	0	0	0									
		lha2	0	0	0	0	0	0									
		lha4	0	0	0	0	0	1									
	lha5	0	0	0	0	0	0										
	lha6	0	0	0	0	0	0										
	lha7	0	0	0	0	0	0										
	lha9	0	0	0	0	0	0										
	lha11	0	0	0	0	0	0										
	lha12	0	0	0	0	0	0										
	lha13	0	0	0	0	0	0										
	lha14	0	0	0	0	0	0										
	lha15	0	0	0	0	0	0										
	lha16	0	0	0	0	0	0										

Table 2. mouse IFN α subtype expression in liver against long-term huIFN α 2 stimulation

Tissue	Gene	huIFNAR mouse mock treatment (reads/sample)					huIFNAR mouse PegIFN α 2 treatment (reads/sample)					
		CA2	CA5	CA6	CA10	CA13	CA1	CA3	CA4	CA7	CA9	CA11
liver (15 week huIFN α 2 treatment)	ifna1	0	0	0	0	0	0	0	0	0	0	0
	ifna2	0	0	0	0	0	0	0	0	0	0	0
	ifna4	0	0	0	0	0	0	0	0	0	0	0
	ifna5	0	0	0	0	0	0	0	0	0	0	0
	ifna6	0	0	0	0	0	0	0	0	0	0	0
	ifna7	0	0	0	0	0	0	0	0	0	0	0
	ifna9	0	0	0	0	0	0	0	0	0	0	0
	ifna11	0	0	0	0	0	0	0	0	0	0	0
	ifna12	0	0	0	0	0	0	0	0	0	0	0
	ifna13	0	0	0	0	0	0	0	0	0	0	0
	ifna14	0	0	0	0	0	0	0	0	0	0	0
	ifna15	0	0	0	0	0	0	0	0	0	0	0
	ifna16	0	0	0	0	0	0	0	0	0	0	0

Lastly, we want to express our heartfelt appreciation again. With your suggestion, our manuscript becomes more solid.

3. The knock-in of the human IFNAR1 and IFNAR2 genes into the mouse IFNAR2 locus must be validated and shown by sequencing.

Response: Thanks for your suggestion. The knock-in sequence is provided in the supplementary information in Appendix 1 in this revision. The sequencing results are also provided in Appendix 2. We hope this two information make our huIFNAR design easy to understand and widely repeatable.

IFNAR2 Cas9-KI Targeted vector sequence information

Recombinant Arms are Red, Knock-in fragment is yellow, exon2 is blue, polyA is gray.

ACCAGA ATTTGG CTA CTT TTTAAT TTATTT TTAAG ACAGGT AGCTCC TCTAGC TGGCTT
CAAAC TGTAT GTAGCC GAGGAT GACCTT GAATGC CTGACC TTTCCC CCACCT CTCTGC TTAGAG
GACAGA TGTGAC ACGCAC AGTAAA CTCATG CAAGTT TAACCC TAATCC TAACCA ATCCAG GGCTAC
CACGGG GCCGCA TCTGCA GCTAAA TCTGGC TCGTTC TTA CTTCT GTCTCT CGTTAG CGTGTA TGTGTC
TATCAT GTAAAT TACAAT ATAATT GGGTGC TTCTGA GTTTTG ACCAAC TCAATA TTGATC TCTTTC
AGGTGT GAGAGC AGAAAA ACGGAC TTAAGA GCTGAG CAGGAT GCTGCT GTCCCA GAATGC CTTCAT
CTTCCG CTCCCT GAACCT GGTGCT GATGGT CTACAT CTCCCT GGTCTT TGGCAT CTCCTA TGACTC
CCCTGA CTACAC AGATGA GTCCTG CACCTT CAAGAT CTCCCT GAGGAA CTTTCA GTCCAT CCTGTC
CTGGGA GCTGAA GAACCA CTCCAT TGTGCC CACCCA CTACAC CCTGCT GTACAC CATCAT GTCCAA
GCCTGA GGACCT GAAGGT GGTGAA GAACTG TGCCAA CACCAC CAGGTC CTTCTG TGACCT GACCGA
TGAGTG GAGGTC CACCCA TGAGGC CTATGT GACAGT GCTGGA GGGCTT CTCTGG CAACAC CACCCT
GTTCTC CTGCTC CCACAA CTTCTG GCTGGC CATTGA CATGTC CTTTGA GCCCCC TGAGTT TGAGAT
TGTGGG CTTTCA CAACCA CATCAA TGTGAT GGTGAA GTTCCC ATCCAT TGTGGA GGAGGA GCTGCA
GTTTGA CCTGTC CCTGGT GATTGA GGAGCA GTCTGA GGGCAT TGTGAA GAAGCA CAAGCC TGAGAT

CAAGGG CAACAT GTCTGG CAACTT CACCTA CATCAT TGACAA GCTGAT CCCCAA CACCAA CTA CTG
TGTCTC TGTCTA CCTGGA GCACTC TGATGA GCAGGC TGTGAT CAAGTC CCCCTT GAAGTG CACCCT
GCTGCC CCCTGG CCAGGA GTCTGA GTCTGC TGAGTC TGCCAT TGTGGG CATCAC CACCTC CTGCCT
GGTGGT GATGGT CTTTGT CTCCAC CATTGT GATGCT GAAGAG GATTGG CTACAT CTGCCT GAAGGA
CAACCT GCCCAA TGTGCT GAACTT CAGGCA CTTCTT GACCTG GATCAT CCCTGA GAGGTC CCCATC
TGAGGC CATTGA CAGGCT GGAGAT CATCCC CACCAA CAAGAA GAAGAG GCTGTG GAACTA TGACTA
TGAGGA TGGCTC TGA CTC TGATGA GGAGGT GCCCAC AGCCTC TGTGAC AGGCTA CACCAT GCATGG
CCTGAC AGGCAA GCCCCT GCAGCA GACCTC TGACAC CTCTGC CTCCCC TGAGGA CCCCTT GCATGA
GGAGGA CTCTGG CGCTGA GGAGTC TGATGA GGCTGG CGCTGG CGCTGG CGCTGA GCCTGA GCTGCC
CACAGA GGCTGG CGCTGG CCCATC TGAGGA CCCCAC AGGCC ATATGA GAGGAG GAAGTC TGTGCT
GGAGGA CTCCTT CCCAG GGAGGA CAACTC CTCAT GGATGA GCCTGG CGACAA CATCAT CTTCAA
TGTGAA CCTGAA CTCTGT CTTCTT GAGGGT GCTGCA TGATGA GGATGC CTCTGA GACCCT GTCCCT
GGAGGA GGACAC CATCCT GCTGGA TGAGGG TCCCCA GAGGAC AGAGTC TGACCT GAGGAT TGCTGG
CGGCGA CAGGAC CCAGCC CCCCTT GCCATC CTTGCC ATCCCA AGACCT GTGGAC AGAGGA TGGCTC
CTCTGA GAAGTC TGACAC CTCTGA CTCTGA TGCTGA TGTGGG CGATGG CTACAT CATGAG GGGATC
CGGCTC TGGCTC TGGCTC TGGCTC TGGCTC TGGCGC CACCAA CTTCTC CTTGCT GAAGCA GGCTGG
CGATGT GGAGGA GAACCC TGGGCC CATGAT GGTGGT GCTGCT GGGCGC CACCAC CTTGGT GCTGGT
GGCTGT GGCCC ATGGGT GCTGTC TGCTGC TGCTGG CGGCAA GAACCT GAAGTC CCCCA GAAGGT
GGAGGT GGACAT CATTGA TGACAA CTTTAT CTTGAG GTGAA CAGGTC TGATGA GTCTGT GGGCAA
TGTGAC CTTCTC CTTTGA CTACCA GAAGAC AGGCAT GGACAA CTGGAT CAAGCT GTCTGG CTGCCA
GAACAT CACCTC CACCAA GTGCAA CTTCTC CTCCTT GAAACT GAATGT CTATGA GGAGAT CAAGCT
GAGGAT CAGGGC TGAGAA GGAGAA CACCTC CTCCTG GTATGA GGTGGA CTCCTT CACCC ATCCG
CAAGGC CCAGAT TGGCCC CCCTGA AGTGCA TCTGGA GGCTGA GGACAA GGCCAT TGTGAT CCACAT
CTCCCC TGGCAC CAAGGA CTCTGT GATGTG GGCTCT GGATGG CTTGTC CTTTAC CTACTC CTTGGT
GATCTG GAAGAA CTCCTC TGGCGT GGAGGA GAGGAT TGAGAA CATCTA CTCAG GCACAA GATCTA
CAAGCT GTCCCC TGAGAC CACCTA CTGCCT GAAGGT GAAGGC TGCCCT GCTGAC CTCCTG GAAGAT
TGGCGT CTACTC CCCTGT GCACTG CATCAA GACCAC AGTGA GAATGA GCTGCC CCCCC TGAGAA
CATTGA GGTCTC TGTGCA GAACCA GAACTA TGTGCT GAAGTG GACTA CACCTA TGCCAA CATGAC
CTTCCA AGTGCA GTGGCT GCATGC CTTCTT GAAGAG GAACCC TGGCAA CCATCT GTACAA GTGGAA
GCAGAT CCCTGA CTGTGA GAATGT GAAGAC CACCCA GTGTGT CTTCCC CCAAAA TGTCTT CCAGAA
GGGCAT CTACCT GCTGAG GGTGCA GGCCTC TGATGG CAACAA CACCTC CTTCTG GTCTGA GGAGAT
CAAGTT TGACAC AGAGAT CCAGGC CTTCTT GCTGCC CCCTGT CTTCAA CATCCG CTCCCT GTCTGA
CTCCTT CCACAT CTACAT TGGCGC CCCCAA GCAGTC TGGCAA CACCC TGTGAT CCAGGA CTACCC
CCTGAT CTATGA GATCAT CTTCTG GGAGAA CACCTC CAATGC TGAGAG GAAGAT CATTGA GAAGAA
GACAGA TGTGAC AGTGCC CAACCT GAAGCC CTTGAC AGTCTA CTGTGT GAAGGC CAGGGC TCACAC
CATGGA TGAGAA GCTGAA CAAGTC CTCTGT CTTCTC TGATGC TGTCTG TGAGAA GACCAA GCCTGG
CTCCTT CTCCAC CATCTG GATCAT CACCGG CTTGGG CGTGGT CTTCTT CTCTGT GATGGT GCTGTA
TGCCCT GAGGTC TGTCTG GAAGTA CTTGTG CCATGT CTGCTT CCCCC CTTGAA ACCCC CCGCTC
CATTGA TGAGTT CTTCTC TGAGCC CCCATC CAAGAA CTTGGT GCTGCT GACAGC TGAGGA GCACAC
AGAGCG CTGCTT CATCAT TGAGAA CACAGA CACAGT GGCTGT GGAGGT GAAGCA TGCCCC TGAGGA
GGACCT GAGGAA GTACTC CTCCCA GACCTC CCAAGA CTCTGG CAACTA CTCCAA TGAGGA GGAGGA
GTCTGT GGGCAC AGAGTC TGGCCA GGCTGT GCTGTC CAAGGC CCCATG TGGCGG CCCATG CTCTGT

```

GCCATC CCCCCC TGGCAC CCTGGA GGATGG CACCTG CTTCTT GGGCAA TGAGAA GTACCT GCAGTC
CCCTGC CCTGAG GACAGA GCCTGC CCTGCT GTGCTA AACTC CTCAGG TGCAGG CTGCCT ATCAGA
AGGTGG TGGCTG GTGTGG CCAATG CCCTGG CTCACA AATACC ACTGAG ATCTTT TTCCCT CTGCCA
AAAATT ATGGGG ACATCA TGAAGC CCCTTG AGCATC TGAATT CTGGCT AATAAA GGAAAT TTATTT
TCATTG CAATAG TGTGTT GGAATT TTTTGT GTCTCT CACTCG GAAGGA CATATG GGAGGG CAAATC
ATTTAA AACATC AGAATG AGTATT TGGTTT AGAGTT TGGCAA CATATG CCCATA TGCTGG CTGCCA
TGAACA AAGGTT GGCTAT AAAGAG GTCATC AGTATA TGAAAC AGCCCC CTGCTG TCCATT CCTTAT
TCCATA GAAAAG CCTTGA CTTGAG GTTAGA TTTTTT TTATAT TTTGTT TTGTGT TATTTT TTTCTT
TAACAT CCCTAA AATTTT CCTTAC ATGTTT TACTAG CCAGAT TTTTCC TCCTCT CCTGAC TACTCC
CAGTCA TAGCTG TCCCTC TTCTCT TATGGA GATCCC TCGACC TGCAGA TGCATT CACGGT GCACAG
TCTCTG CCGTCG GTCTCC TCAGCT TGTGTC TTGTGG GTAAGG GCTACT TCTCAG CACAGC CCTTAG
AGGAGA AAGCCT CTGTTT CTGTCA TCACAG AGAGCC CTGGTG TGGAGC AGCACA CTGATG TCCATA
TCTGGA GAACCC AGATCA GCACGG CCAGCA TCAGGC ACCCCA CGGGGG TCTTCC CCTTCA TTTTAG
CTAAGC CAGAAT AATATA GGCTAC AGCCAT ATTCTG GAAACG GCCTTG TTTATA ATTCAG AGGGTT
GCGGCT CTGCAC ACCCTG AATCTC ACGCCC GGTGGC GTTTAG AAGGTG GCCATC CCTTTA TCTCTT
CCCATA TAAACT AACTTG AAAAAT CCATCC CTACAC ATTGAT TTATAC TCTTCC TTTCTT

```

4. The mouse genotyping shown in Fig S3 is confusing (and the figure legend is wrong in describing the Hetero: "Hetero, homozygous C57BL/6J-huIFNAR mouse"): why does the Homo show identical amplification with primer pairs #1 and #2 when they should be of different sizes like shown for the Hetero?

Response: Thanks for your comments. During the initial submission, the simple version of a schematic illustration of genotypes for wildtype, heterozygous and homozygous might bring some difficulty in understanding the whole design. Per your suggestion, we have replaced it with a more detailed illustration (as follows). Now, it is clearly shown that only the F3+R3 primer can amplify one 392bp amplicon in the WT mouse. The F3+R3 primer can amplify a far-long amplicon of 4359bp in the homozygous mouse. F1+F2 and F2+R2 can amplify two segments (2975bp and 3168bp, respectively) in both heterozygous and homozygous mice. Due to the heterozygous mouse carrying both the wildtype and huIFNAR-inserted alleles, the heterozygous mouse has one more band (392bp) than the homozygous mouse.

We hope the current illustrations are clear.

Figure 3 for review : An overview illustration of genotyping strategy.

Homologous recombination arms at the 5' and 3' are Knock in fragment are highlighted. Primer sites and product sizes are labelled.

5. Fig 1D: what cells were used to analyze hIFNAR expression by FACS? Why are all the cells positive? Is there residual mouse IFNAR expression? Show the knock-in is complete by FACS. What is the residual mouse IFNAR response in the HuIFNAR mice when stimulated with mouse IFN α ?

Response: Thanks for your comments.

1) In Fig 1D, B cells were used for the FACS analysis. We are using splenic cells for analysis. The procedure to isolate single cells from the spleen is easy, and B cells are abundant. Since we are using the endogenous mouse *lfnar2* promoter, the huIFNAR will be expressed and function in an authentic tissue-specific manner to their counterpart in wildtype mice. Since B cells express IFNAR2, it's no surprise that all B cells are positive to be stained in Fig 1D.

2) We used the knock-in strategy to generate the huIFNAR mouse by inserting the whole humanized CDS plus transcription termination signal into the mouse *lfn2* exon. Since humanized IFNAR2+R1 tandem CDS is site-specifically integrated into the endogenous gene body of mouse *lfnar2*, the expression of the inserted huIFNAR2+R1 gene will follow an identical manner to the endogenous mouse *lfnar2* gene.

3) Mouse *Ifnar1* gene knockout results in the loss of function of the type I Interferon pathway and more susceptible to infection. In the widely used B6.129S2-*Ifnar1*^{tm1Agt/Mmjax} mice (for short A129 mice, provided by Jax lab, <https://www.jax.org/strain/010830>), homozygous for this targeted allele (*Ifnar1*^{-/-}) lack type I IFN receptor function, exhibit enhanced osteoclastogenesis with decreased bone density, and impaired response to protozoan parasite (*Leishmania*) infection. The *Ifnar1*^{-/-} mice are susceptible to HAV (*Science*. 2016 Sep 30;353(6307):1541-1545. doi: 10.1126/science.aaf8325) and Dengue virus infection (*Nat Commun*. 2017 Nov 13;8(1):1459. doi: 10.1038/s41467-017-01669-z.).

However, in our study, the hIFNAR mice still have a weak response to mouse IFN α 5 stimulation (supplementary figure 6 in this revision). We guess, the residual mouse IFNAR1 might still form dimmers to respond non-specifically to mIFN α 5 stimulation. The RNA sequencing result is more sensitive than FACS staining analysis. Therefore, we don't include a FACS analysis.

Your valuable comments make us determined to further knockout the mouse IFNAR1 on top of the current hIFNAR mouse. We have started to generate this mouse recently. But it will take at least one year to get a workable colony of homozygous hIFNAR and mouse IFNAR1 knockout mice to test this postulation.

6. The functional assessment of hIFN α 2 responses in the spleen shown in Fig 1I and 1J is problematic: mMX1 expression was present in only 2/4 mice and mISG15 0/4 mice. How do the authors explain this lack of response in the spleen? It seems unlikely that there is any statistical significance Fig 1J in the spleen ($p=0.0369$), as there is no expression! The text is not truthfully describing the results.

Response: Thanks for your comment. Other reviewers also questioned this part. We repeated the experiment and have made substantial changes to Figure 1. All the mice responded to IFN α stimulation, and statistical differences were labelled correspondingly. This part is rephrased in the main text.

7. It is not clear why the GO category "response to type I interferon" is close to zero in Fig 2C although these cells were stimulated with PEG-IFN α 2, can the authors explain why?

Fig S5A: most of the genes are illegible.

Response: Thanks for your interest in the GO analysis. As you know, we have changed Figure 2 as requested by all three reviewers. In this revision, we stimulated the mouse PBMC in vitro instead. After analysis, the top 30 GO categories did not include “response to type I interferon”.

FigS5A is now (as Fig. S7) in a high-resolution PDF file and is easy to read.

8. Fig 3C the gene names must be aligned with the heatmap, or else we can not analyze the data.

Response: Thanks for your comment. There are too many genes in each lane. For the limited space, it is hard to label all the gene names just beside the figure. Now, the figure is in high resolution, and gene names (45 genes) are well-labelled. We hope you can easily read the tissue-specific gene expression patterns.

9. The AAV-1.3xHBV genotype C injected mice show a profound drop in HBsAg levels at week 2-3 which is not paralleled by the HBeAg nor the HBV DNA levels (Fig S7). This is very uncommon and has not been described by other groups. Please explain.

Response: It's interesting that a profound drop of HBsAg, but not HBe and HBV DNA, occurs at week 2-3 post AAV-HBV 1.3X genotype C injection. We have repeated and confirmed this phenomenon multiple times.

Our explanations are the following.

1) Genotype might account for the drop. We are using an HBV C2 genotype strain isolated from a Chinese patient. The HBV C2 genotype strains are popular in China. We also generate the AAV-HBV B genotype. Genotype B doesn't have an HBsAg drop.

2) Acute infection precludes chronic infection. Chronic HBV infection usually includes acute and chronic phases. In a typical acute HBV infection, the host will suppress virus replication through an unknown mechanism (reference: N Engl J Med 2004;350:1118-29: Hepatitis B Virus Infection — Natural History and Clinical Consequences). If the host fails to clear HBV, HBV infection will become chronic. The HBV C2 genotype mouse model mimics HBV infection in vivo.

3) A relatively limited representativeness of AAV-HBV to the authentic HBV cccDNA. AAV vector serves as a trojan horse to deliver a linear form of the HBV genome to the nuclear.

In the circular HBV cccDNA template, only four promoters regulate the 5 HBV mRNA transcription from the overlapped HBV cccDNA. The cccDNA can transform into supercoiled mini-chromosome. These 4 promoters might affect each other in a well-coordinated and synergistic manner. For example, the inhibition of promoters 2 and 3 might also directly inhibit the downstream promoter 4 and promoter 1, which control the HBx and PreCore and Core mRNA transcription. The drop of HBsAg will likely result in the decline of the HBx and HBx and HBV DNA levels.

HBV gene expression from HBV cccDNA

Figure 4 for review : HBV viral mRNA transcription and protein expression from HBV cccDNA template. Promoter 1, 2, 3 and 4 are indicated.

However, the AAV-HBV 1.3-fold genome vector contains six promoters, consisting of two promoter 1, one promoter 2, one promoter 3 and two promoter 4. In the same circumstance when promoters 2 and 3 are inhibited, the promoters 4 and 1 downstream

(within the 1.0X HBV genome) might be suppressed too. But, the other promoters 4 and 1 upstream (within the 0.3X genome) might be still active and still be able to transcribe the HBx mRNA (0.7kb) and preCore and pre-genomic RNA (3.5 kb). Therefore, the HBsAg drop was not accompanied by an HBeAg and HBV DNA decline in the AAV-HBV 1.3X model.

There are still other models to explain the difference between AAV-HBV 1.3X and HBV cccDNA.

HBV gene expression from AAV-HBV 1.3X

Figure 5 for review : HBV viral mRNA transcription and protein expression from AAV-HBV 1.3X template. Promoter 1, 2, 3 and 4 are indicated.

10. How do the viral kinetics in huIFNAR mice compare to wild type mice? This control is necessary. To properly evaluate the effect of PEG-IFNa treatment in the huIFNAR AAV-1.3xHBV mice we need to see the viral kinetics prior to the PEG-IFNa treatment in Fig 4 (-6w to 0w).

Response: Thanks for your comments.

To address this concern, we put all the information in the supplementary due to the limited space in Figure 4. We provide a **supplementary Figure 10** to show the HBV viral kinetics in huFNAR and wildtype mice. The HBsAg levels in huFNAR mice were higher than in wildtype mice at week 2 post AAV-HBV injection, but their levels became equivalent at week 6. The HBeAg levels also showed a difference at week 2, but became comparable at week 6. That is why we chose week 6 as the starting point for IFNa-treatment.

The HBV DNA level prior to IFNa treatment is added in the supplementary figure 12.

I hope it will be helpful to understand the new model.

Supp. Fig. 10

Figure 6 for review : Chronic HBV infection HuIFNAR AAV-HBV mouse model. (A)

Experimental protocol. HuIFNAR mouse model of chronic AAV-HBV infection was established by tail vein injection of a 1.3-fold HBV genome (AAV-1.3×HBV). After that, Peg-IFNα2 was injected. (B.C) Kinetics of HBsAg (IU/ml) and HBeAg (IU/ml) of each mouse prior to the IFN α 2 treatment. Data in are means ± SEM, p values determined by one-way ANOVA.

11. The HBeAg data needs to be revised to match between Fig 4 (IU/ml) and Fig S7 (OD450).

Response: We have been working on AAV-HBV mouse model for over five years. When the lab was just set up in a new location, we didn't have standard HBeAg (determined by IU) to measure serum HBeAg concentration. We had to use OD450 instead at that time. As you know, we only could have less than 50ul serum from mice by the mouse ethic committee. We also need to measure multiple biomarkers and detection. Unfortunately, we already ran out of samples several years ago. Therefore, we are sorry for being unable to retest the HBeAg concentration. Please allow us to keep the HBeAg in the OD450 value (now in Supplementary Figure 9). We will appreciate your kindness.

12. What are the HBV DNA kinetics throughout the experiment (show before IFN α treatment)?

Response: Thanks for your comment.

The kinetics of HBsAg, HBeAg, and HBV DNA prior to and post human IFN treatment are included in Supplementary Figures 10 and 12. Please also refer to the response to comment 10.

In addition, in the current study, we are using this model to address whether human IFN treatment could inhibit HBV, particularly the HBsAg. Meanwhile, AAV-HBV has been extensively used in the fields for modelling HBV infection. Therefore, we are focusing on testing the efficacy of human IFN on HBsAg expression.

13. Fig S9 shows that the groups are biased: the WT mice start with higher levels of HBV DNA than the Mock or IFN α 2 treated mice. The bias in the experimental groups is also visible in Fig 4J: the WT mice have a higher level of intrahepatic HBV DNA than the Mock or IFN α groups. Why do the huIFNAR mice have lower HBV DNA levels than controls?

Response: Thanks for your valuable comments.

This observation is the most un-explainable one in our study. The same dose of AAV-HBV (2×10^{11} vg/mouse) was injected into the tail vein of C57 wildtype and huIFNAR mice, respectively. We observed equal levels of HBsAg and HBeAg in both groups. However, the HBV DNA levels in the huIFNAR group were markedly low.

We postulate that AAV-HBV is not the authentic representative HBV model, although AAV-HBV is currently the most representative mouse model. Please be patient to hear our postulation. We have shown our explanation above. Please refer to the response to comment 9 for the details.

14. Seroconversion of HBs has been described in AAV-1.3xHBV mice since the mouse immune response is fully competent to develop humoral immunity (Huang YH et al. Int J Oncol 2011). Thus mouse #3 is not uncommon and could occur without hulfNAR if larger cohorts of animals are used.

Response: Your comment that “Seroconversion of HBs has been described in AAV-1.3xHBV mice since the mouse immune response is fully competent to develop humoral immunity” really surprised us. I download the literature and read it with great interest.

Reference 1: Int J Oncol. 2011 Dec;39(6):1511-9. doi: 10.3892/ijo.2011.1145. Epub 2011 Jul 28. A murine model of hepatitis B-associated hepatocellular carcinoma generated by adeno-associated virus-mediated gene delivery Ya-Hui Huang, Cheng-Chieh Fang, Koichi Tsuneyama, Ho-Yuan Chou, Wen-Yu Pan, Yao-Ming Shih, Ping-Yi Wu, Yin Chen, Patrick S C Leung, M Eric Gershwin, Mi-Hua Tao Cited by 18 times from Pubmed database.

Unfortunately, their study used a recombinant version of AAV-HBV vector combination, consisting of AAV8/5'-HBV-SD and AAV8/3'-HBV-SA (**Reference 1**). This recombinant version of the AAV-HBV model has been referred to only 18 times in the Pubmed database and has been used less in recent publications.

This recombinant version of the AAV-HBV mouse model has the following limits,

1) **High cost.** AAV-HBV preparation is costly and technically challenging. Achieving 10^6 copies/ml HBV virus in the blood required the injection of over 10^{12} AAV-HBV (viral particles)/mouse. Two split AAV-HBV vectors (AAV8/5'-HBV-SD and AAV8/3'-HBV-SA) are required.

[REDACTED]

Figure 8 for review : Figure from Int J Oncol. 2011 Dec;39(6):1511-9. doi:

10.3892/ijo.2011.1145. Epub 2011 Jul 28. A murine model of hepatitis B-associated hepatocellular carcinoma generated by adeno-associated virus-mediated gene delivery

2) **No HBsAg seroconversion spontaneously occurred.** In the literature, we could not find HBsAg seroconversion (HBsAg loss and HBsAb generation) among the 12 AAV-HBV mice at 12 to 16 months post-injection. Also, in the main text, the authors clearly stated that **significant amounts of HBs (mean titer 2541 IU/ml) were detected in mice transduced with AAV/HBV.**

[REDACTED]

Most importantly, 10 of 12 mouse developed HCC.

[REDACTED]

In addition, although the mouse has full immune competence, they are unlikely to get HBsAg loss in the short-term, such as within only several weeks.

I want to explain briefly about the AAV-HBV 1.3-fold mouse model widely used in the HBV fields. The virtues of this model are the following.

1) One AAV vector is enough. AAV-HBV 1.3-fold genome, which carries an extra 0.3-fold HBV genome as the preCore and Core promoter, is widely used to generate the AAV-HBV mouse model (as illustrated in the following). One AAV vector is functionally similar to HBV cccDNA in vivo in producing HBV viral full particles (Dane particle, containing HBV genomic DNA and infectious) and sub-viral particles (without HBV genomic DNA and non-infectious).

HBV gene expression from AAV-HBV 1.3X

Figure 9 for review : HBV viral mRNA transcription and protein expression from AAV-HBV 1.3-fold genomic DNA template. Promoter 1, 2, 3 and 4 are indicated.

2) Stable and non-carcinogenic.

OF NOTE. AAV-HBV has been extensively used for modelling chronic HBV infection in mice. We randomly select five papers using AAV-HBV models, as the following.

Reference 1. J Hepatol. 2021 Nov;75(5):1072-1082. doi: 10.1016/j.jhep.2021.06.038. Epub 2021 Jul 7. Novel function of SART1 in HNF4 α transcriptional regulation contributes to its antiviral role during HBV infection

Reference 2. JHEP Rep . 2022 Jul 9;4(9):100535. doi: 10.1016/j.jhepr.2022.100535. eCollection 2022 Sep. Molecular clones of genetically distinct hepatitis B virus genotypes reveal distinct host and drug treatment responses

Reference 3. Nat Nanotechnol . 2020 May;15(5):406-416. doi: 10.1038/s41565-020-0648-y. Epub 2020 Mar 2. Dual-targeting nanoparticle vaccine elicits a therapeutic antibody response against chronic hepatitis B

Reference 4. J Hepatol. 2023 Apr;78(4):717-730. doi: 10.1016/j.jhep.2022.12.013. Epub 2023 Jan 9. Activation of CD4 T cells during prime immunization determines the success of a therapeutic hepatitis B vaccine in HBV-carrier mouse models

Reference 5. Gut. 2023 Aug;72(8):1544-1554. doi: 10.1136/gutjnl-2022-327059. Epub 2022 Oct 31. Engineered anti-PDL1 with IFN α targets both immunoinhibitory and activating signals in the liver to break HBV immune tolerance

Liguo Zhang, the corresponding author of this manuscript published one AAV-HBV mouse model using the AAV-HBV 1.3X in 2014. This paper has been widely cited in the fields.

Reference 2: Cell Mol Immunol . 2014 Jan;11(1):71-8. doi: 10.1038/cmi.2013.43. Epub 2013 Sep 30. A mouse model for HBV immunotolerance and immunotherapy Dan Yang, Longchao Liu, Danming Zhu, Hua Peng, Lishan Su, Yang-Xin Fu, Liguo Zhang

Cited by 52 times from Pubmed database.

We followed the HBV titer up to 57 weeks. We do not observe HBV DNA decline or any HCC in the mouse.

Figure 10 for review: Kinetics of serum HBV DNA in AAV-HBV mouse model. Y-axis, HBV DNA log10 copies/ml. X-axis, weeks post AAV-HBV injection.

Therefore, HBsAg spontaneous seroconversion is extremely, extremely rare if it occurs. It's too imprudent to conclude that HBsAg loss in the AAV-HBV mouse.

15. Figure and legends are inverted for Fig S8 and Fig S9.

Response: We have corrected the mislabel. Thanks.

16. Text needs major revision, page 14 line 299 “As IFNs do not directly act on the intracellular HBV viral genome” this is false and forgetting some important articles: Zhang M et al. J Hepatol 2021, Baudi I et al. Plos Pathogens 2021, Allweiss L et al. Gut 2022.

Response: We sincerely appreciate your keen attention to detail in pointing out the incorrect description. The three references are also included in this revision. Now, we have rephrased the first sentence to be the following.

Human IFNa2 seems to inhibit HBV replication in multiple ways^{17,25-27}.

17. Overinterpretation of the data from two mice in Fig S10D: no statistics are possible, the authors can not state page 15 line 310-312 “the PEG-IFNa2 treatment substantially

increased...". This is the same issue in Fig 5B: only two mice were analyzed and the text describes these results as significant, although no statistics can be run.

Response: We sincerely appreciate your keen attention to detail in pointing out the over-interpretation in these two instances. To ensure precision, we have replaced the terms "substantially" or "significantly" with the phrase "IFNa treatment showed a tendency to increase."

Regarding the methodology employed in our study, we would like to clarify that due to the challenge of obtaining an adequate number of liver leukocytes for multiple analyses, including two additional FACS (Figure 7) and ELISPOT (results not shown in the manuscript but provided for review purposes), we implemented a sample-pooling strategy for the single-cell sequencing analysis. Pooling samples for single-cell sequencing analysis is common practice when the available cell numbers are limited. Specifically, in Figure S10D and Fig.5B, each dot represents pooled cells from two to three mice. Thus, the two dots in each column represent all the 5 to 6 mice in the experimental group.

We hope you understand the challenges we faced in this regard. Additionally, it is important to note that single-cell sequencing analysis incurs substantial costs in our laboratory. Conducting extensive analysis on a larger number of samples at multiple time points would require additional grant support. We kindly request your support and understanding in facilitating our grant application, which would enable us to conduct more comprehensive analyses in the future. Your generosity in this matter would be greatly appreciated.

19. Fig 7. What are the control mice? How was the liver prepared for the intrahepatic leukocyte analysis? Liver resident macrophages need to be stained with CD68. Proinflammatory macrophages are mostly in the circulating blood. Expression of PD-1 on CD8+ T cells does not necessarily mean the cells are exhausted since PD-1 is upregulated on activated cells.

Response: Thanks for your comment.

1) In Figure 7, the wildtype C57/BL6 AAV-HBV mice treated with IFNa was used as control mice. Your comment reminds us that the word "control", inconsistently used throughout the manuscript, will confuse the audience. Therefore, we re-labelled the groups in Figure 7 in a manner consistent with what was used in Figure 4. To be easily read, WT-IFNa2, huIFNAR-Mock, and huIFNAR-IFNa2 are appropriately marked on the top of each small figure in Figure 7.

2) In this revision, we have included the preparation of intrahepatic leukocytes in the methods part.

Intrahepatic lymphocyte single cell isolation

Intrahepatic lymphocyte cell isolation was done by enzymatic digestion. Briefly, liver was minced into 0.5-3mm pieces in digestion buffer (RMPI 1640 + 5%FBS + 20ng/ml DNase I + 0.5mg/mL Collagenase D), and enzymatically digested with the program 37C_m_LIDK_1 by gentleMACS (Miltenyi Biotec, Germany). Liver homogenates were passed through a 70- μ m cell strainer (BD, USA) and then centrifuged at 300rpm for 5 min. Resuspend the pellet in 5 ml of 40% Percoll and centrifuged at 2000rpm for 20 min without brake. After the supernatant was removed, the pelleted cells were suspended in RBS lysis buffer and incubated on ice for 5 min to lyse red blood cells. After washing twice with PBS, the cell pellets were re-suspended in PBS (containing 0.04%BSA).

3) As for the CD68 staining, we had thought to use CD68 to stain the liver resident macrophages. Unfortunately, we didn't get a CD68 FACs staining antibody with a suitable dye in the 21-color staining protocol. Therefore, we use Ly6c+ and CD11b to gate the proinflammatory macrophage. We are sorry for not being able to get the right antibody during the SARS-CoV-2 pandemic.

4) We acknowledge and agree with your comment that the expression of PD-1 on CD8+ T cells does not necessarily indicate cellular exhaustion, as PD-1 can also be upregulated on activated cells. In Figure 6, we observed increased expression of Lag3, Havcr2, Ctla4, and PD-1 on CD8 cells, all commonly used to define cellular exhaustion. Figure 7 serves to corroborate the findings presented in Figure 6. Based on these observations, we concluded that elevated PD-1 expression predominantly represents cellular exhaustion rather than activation (since activation does not necessarily imply loss of function). The discussion section further addresses the exhaustion status of CD8 T cells.

Considering this feedback, we retain the original description that elevated PD-1 expression signifies exhaustion in this revised version.

Materials and Methods

- Remove the chapter "Study design" which has no reason to be in the manuscript.

Response: Thanks for your comment. This paragraph is deleted in this revision.

- Plasmid construction details must be included, "Details will be provided upon request" is not acceptable.

Response: Thanks for your comment. We have provided the plasmid sequence information as Appendix 1 in the supplementary information in this revision. We hope the detailed information will benefit the understanding of this novel mouse and also make the mouse reproducible among different labs.

We also provided the mouse genomic DNA sequencing information as Appendix 2, confirming that we got the right mouse.

- The preparation of the liver leukocytes is missing.

Response: Thanks for your comment. We did miss the preparation of the liver leukocytes in the methods part during the initial submission. We have included the methods in this revision.

Intrahepatic lymphocyte single cell isolation

Intrahepatic lymphocyte cell isolation was done by enzymatic digestion. Briefly, liver was minced into 0.5-3mm pieces in digestion buffer (RMPI 1640 + 5%FBS + 20ng/ml DNase I + 0.5mg/mL Collagenase D), and enzymatically digested with the program 37C_m_LIDK_1 by gentleMACS (Miltenyi Biotec, Germany). Liver homogenates were passed through a 70- μ m cell strainer (BD, USA) and then centrifuged at 300rpm for 5 min. Resuspend the pellet in 5 ml of 40% Percoll and centrifuged at 2000rpm for 20 min without brake. After the supernatant was removed, the pelleted cells were suspended in RBS lysis buffer and incubated on ice for 5 min to lyse red blood cells. After washing twice with PBS, the cell pellets were re-suspended in PBS (containing 0.04%BSA).

REVIEWERS' COMMENTS

Reviewer #1 (Remarks to the Author):

Thank you for the revision of the manuscript. I am satisfied with the answers and endorse the publication of the manuscript.

Reviewer #3 (Remarks to the Author):

Major comment:

The new information that mIFN α 5 induces responses in the huIFNAR mouse is important regarding the species specificity of the IFN α response in this model. The description of results shown in Fig S6A and S6B needs to be more detailed in the main text. Importantly, the caveat of the model regarding response via the mouse IFNAR1 must be clearly explained in the discussion.

Minor comments:

Abstract line 46-47: "we demonstrated that human IFN stimulated gene expression profiles of huIFNAR peripheral blood mononuclear cells (PBMCs) were similar to that in..."

Fig 1D: need to indicate antibody on x axis.

Furthermore, the title "PegIFN α 2 stimulates identical immune responses in the huIFNAR mouse PBMCs to those in human PBMCs" is misleading and needs to be toned down.

Fig S14. Add to the legend description the color coding for the red dot in the IFN α 2 group.

Line 331 need to revise the sentence "myeloid cells such as natural killer (NK) cells, neutrophils and macrophages". NK cells are of the lymphoid, not myeloid lineage. Please define which cells are the MKi67+ monocytes.

The nomenclature should be revised to be simpler for the reader: "WT-IFN α , huIFNAR-Mock, and huIFNAR-IFN α " should be used throughout the legend figures and text.

REVIEWERS' COMMENTS

Reviewer #1 (Remarks to the Author):

Thank you for the revision of the manuscript. I am satisfied with the answers and endorse the publication of the manuscript.

Response: We express our heartfelt gratitude for your recognition of our study. Thanks.

Reviewer #3 (Remarks to the Author):

Major comment:

1. The new information that mIFN α 5 induces responses in the huIFNAR mouse is important regarding the species specificity of the IFN α response in this model. The description of results shown in Fig S6A and S6B needs to be more detailed in the main text.

Response: We appreciate your comment and agree that more information should be added to improve the clarity. We have added some detail expression about the Fig.s6a and s6b in the main text, as the follow.

Given the species specificity of the type I interferon pathway, we hypothesized that the huIFNAR mouse would have an altered response to human IFN stimulation, compared with the wildtype C57BL/6/C to the mouse IFNs. To test this hypothesis, we compared the gene expression profiles against interferon stimulation in the wildtype and huIFNAR mouse. PBMCs isolated from wildtype C57BL/6C and huIFNAR mice were stimulated with huIFN α 2 and mouse IFN α 5 for 8 h (short-term) and 20 h (long-term), respectively, and subjected to a transcriptomic analysis of the gene expression (Fig. S6a). As expected, the heatmap, containing all the differently expressed genes at different time points post IFN stimulation, showed obvious gene

expression differences between the mIFN α 5 and huIFN α 2 (Fig. S6b), confirming that the huIFNAR mouse altered its type I interferon pathway. Taken together, we successfully generated functional type I interferon receptor-humanized mice.

2. Importantly, the caveat of the model regarding response via the mouse IFNAR1 must be clearly explained in the discussion.

Response: Your reminding is very constructive. We have added one point in the discussion of the limits of our study (as follow). Thanks.

Third, the still kept mouse *Ifnar1* in the current huIFNAR mouse might bring side-effect if used to test mouse IFN in this model. We are generating a new version of the huIFNAR mouse without the mouse *Ifnar1* gene.

Minor comments:

1. Abstract line 46-47: “we demonstrated that human IFN stimulated gene expression profiles of huIFNAR peripheral blood mononuclear cells (PBMCs) were similar to that in...”

Response: This sentence is rephased, as below. Thanks.

... we demonstrate that human IFN stimulates gene expression profiles in huIFNAR peripheral blood mononuclear cells (PBMCs) are similar to those in human PBMCs,

2. Fig 1D: need to indicate antibody on x axis.

Response: Thanks for the comment. We have added “Apc-Streptavidin” on x axis of Fig. 1d.

3. Furthermore, the title “PegIFN α 2 stimulates identical immune responses in the huIFNAR mouse PBMCs to those in human PBMCs” is misleading and needs to be toned down.

Response: Thanks for your reminding. We used a weak tone in this revision “PegIFN α 2 stimulates **similar** immune responses in the huIFNAR mouse PBMCs to those in human PBMCs”.

4. Fig S14. Add to the legend description the color coding for the red dot in the IFN α 2 group.

Response: Thanks for your suggestion. We have added “the dots in red represent mouse #3 in the huIFNAR-IFN α 2 group” in the figure legend of S14.

5. Line 331 need to revise the sentence “myeloid cells such as natural killer (NK) cells, neutrophils and macrophages”. NK cells are of the lymphoid, not myeloid lineage.

Response: Sorry for the mistake. We have made

.....consisting of lymphoid cells such as B cells, T cells, CD8⁺ T cells, CD4⁺ T cells, regulatory T **and natural killer (NK)** cells, and myeloid cells such as **neutrophils and macrophages** (Fig. S15b-e).....

6. Please define which cells are the MKi67⁺ monocytes.

Response: This cell population is defined at the first appearance as MKi67⁺ monocytes (**expressing high Ki-67**).

7. The nomenclature should be revised to be simpler for the reader: “WT-IFN α , huIFNAR-Mock, and huIFNAR-IFN α ” should be used throughout the legend figures and text.

Response: Thanks for your comment. We have used changed the mouse group information to a uniform nomenclature from Figure 4 to 8 appropriately. In those small panels, we also indicate in the figure legend as “ WT, Mock and IFN α 2 represent WT-IFN α 2, huIFNAR-Mock and huIFNAR-IFN α 2, respectively.”

We also confirm other nomenclatures, such as IFN α 2 and IFN α 2a, huIFNAR and HuIFNAR, etc, to be consistent in the manuscript.